# Ca$^{2+}$-regulated Ca$^{2+}$ channels with an RCK gating ring control plant symbiotic associations

Sunghoon Kim [1,2], Weizhong Zeng[1,2], Shane Bernard[3], Jun Liao [4], Muthusubramanian Venkateshwaran[5], Jean-Michel Ane[3] & Youxing Jiang [1,2]

A family of plant nuclear ion channels, including DMI1 (Does not Make Infections 1) and its homologs CASTOR and POLLUX, are required for the establishment of legume-microbe symbioses by generating nuclear and perinuclear Ca$^{2+}$ spiking. Here we show that CASTOR from *Lotus japonicus* is a highly selective Ca$^{2+}$ channel whose activation requires cytosolic/nucleosolic Ca$^{2+}$, contrary to the previous suggestion of it being a K$^+$ channel. Structurally, the cytosolic/nucleosolic ligand-binding soluble region of CASTOR contains two tandem RCK (Regulator of Conductance for K$^+$) domains, and four subunits assemble into the gating ring architecture, similar to that of large conductance, Ca$^{2+}$-gated K$^+$ (BK) channels despite the lack of sequence similarity. Multiple ion binding sites are clustered at two locations within each subunit, and three of them are identified to be Ca$^{2+}$ sites. Our in vitro and in vivo assays also demonstrate the importance of these gating-ring Ca$^{2+}$ binding sites to the physiological function of CASTOR as well as DMI1.

[1] Howard Hughes Medical Institute and Department of Physiology, University of Texas Southwestern Medical Center, Dallas, TX, USA. [2] Department of Biophysics, University of Texas Southwestern Medical Center, Dallas, TX, USA. [3] Department of Bacteriology, University of Wisconsin-Madison, Madison, WI, USA. [4] School of Life Science and Technology, ShanghaiTech University, 201210 Shanghai, China. [5] School of Agriculture, University of Wisconsin-Platteville, Platteville, WI, USA. Correspondence and requests for materials should be addressed to Y.J. (email: youxing.jiang@utsouthwestern.edu)

Symbiotic associations between plants and microbes such as arbuscular mycorrhizal (AM) fungi and nitrogen-fixing rhizobia facilitate the supply of nutrients to their host plants. The AM symbiosis probably appeared with the first land plants[1] and the rhizobium–legume symbiosis is believed to be derived, at least in part, from the ancient AM symbiosis[2–5]. This evolutionary history explains how these two endosymbioses share the same common symbiotic signaling pathway in legumes. Genetic screenings and mutation studies have identified a set of common critical components in the symbiotic signaling pathway using the two representative legumes, *Medicago truncatula* and *Lotus japonicus* as model systems[1,3,5]. Among them, Does not Make Infection 1 (DMI1) from *M. truncatula* has been recognized as a cation channel that resides in the nuclear envelope and is directly involved in generating oscillations of the nuclear and perinuclear concentration of $Ca^{2+}$ often referred to as $Ca^{2+}$ spiking[6–10]. $Ca^{2+}$ spiking is a key process in the early stage of symbiotic signal transduction that results in the regulation of downstream symbiosis-related gene expression. Two DMI1 homologs, named POLLUX and CASTOR, were identified in *L. japonicus* and both of them are required to initiate symbiosis[11]. *L. japonicus* POLLUX is a close homolog of *M. truncatula* DMI1 whereas *L. japonicus* CASTOR (LjCASTOR) is more distantly related. A closer homolog of CASTOR was identified in *M. truncatula*, but it does not seem to play a significant role in symbiosis[12]. Using mutant rescue assays, it was demonstrated that *M. truncatula* DMI1 is sufficient to replace both POLLUX and CASTOR in *L. japonicus* and that reciprocally both POLLUX and CASTOR are required to replace DMI1 in *M. truncatula*. This functional difference between DMI1 and CASTOR/POLLUX was attributed to a Ser-to-Ala substitution in the filter region of DMI1[12]. Interestingly, expressing DMI1 in human embryonic kidney (HEK) 293 cells allows cytosolic oscillations of the $Ca^{2+}$ concentration reminiscent of $Ca^{2+}$ spiking observed in plant cells[12,13].

CASTOR, POLLUX, and DMI1 channels were predicted to contain four transmembrane (TM) segments followed by a sizeable soluble domain. Sequence analysis suggested that the last two TM regions form the pore and the N-terminal half of the soluble domain is distantly homologous to a Regulator of $K^+$ Conductance (RCK) domain[6,11,12,14]. However, the second half of the soluble domain has no sequence homology to any known proteins. The RCK domain is commonly found in a majority of prokaryotic ligand-gated $K^+$ channels[15–19] and eukaryotic slo family $K^+$ channels such as large conductance, $Ca^{2+}$-gated $K^+$ (BK) channel[20–23]. Initial studies using yeast rescue assays and artificial lipid bilayers suggested that CASTOR, POLLUX, and DMI1 could be $K^+$ channels[12,24] with the RCK-containing MthK channel used as a homology model for a part of these channels (the last two TM regions and the first half of the soluble domain)[11]. Recently, DMI1 was found to interact with $Ca^{2+}$-permeable cyclic nucleotide-gated (CNGC) channels at the nuclear envelopes[25]. One current model of the early symbiotic signaling pathway speculates that DMI1, CASTOR, and POLLUX function as ligand-gated $K^+$ channels that can modulate the membrane potential across the nuclear envelope and that they work together with CNGC channels to modulate nuclear $Ca^{2+}$ release[12,25].

However, several important issues prompted us to revisit the $K^+$ channel model for CASTOR, POLLUX, and DMI1. Firstly, in artificial planar lipid bilayer assays, denatured DMI1 or CASTOR proteins were used in proteoliposome reconstitution[12,24], raising concerns that these channels may not be properly folded and reflect function in cells; up to now, there are no direct electrophysiological measurements of these channels in a cell-based expression system. Secondly, the

predicted pore of these nuclear channels lacks the TVGYG signature sequence of the selectivity filter that is highly conserved in canonical $K^+$ channels. Thirdly, due to low sequence similarity, whether the soluble domain of these channels contains an RCK domain remains to be proven structurally. Furthermore, if the soluble domain of these channels serves as a ligand binding domain similar to other RCK-regulated channels, the functional ligand remains to be identified.

To address these fundamental issues that define the primary function of these channels and their role in the symbiotic process, we performed functional characterization of the LjCASTOR channel and demonstrate that it functions as a $Ca^{2+}$-regulated $Ca^{2+}$ channel rather than as a $K^+$ channel as previously proposed. We also determined the high-resolution crystal structure of the ligand-binding soluble domain of LjCASTOR and reveal that the entire soluble domain consists of two tandem RCK domains that assemble into a tetrameric gating ring where $Ca^{2+}$ can bind and regulate the channel. As CASTOR, POLLUX, and DMI1 are localized in both the outer and the inner membranes of the nuclear envelope, with their soluble domains facing the cytosol and nucleosol, respectively[8], these channels could directly mediate $Ca^{2+}$ release from the nuclear envelope using their $Ca^{2+}$-binding RCK gating rings to regulate the channel activity by sensing the cytosolic/nucleosolic $Ca^{2+}$ concentrations.

## Results

**CASTOR is a $Ca^{2+}$-regulated $Ca^{2+}$ channel.** Our initial attempt to measure the channel activity by patching the plasma membrane of HEK 293 cells expressing full-length LjCASTOR was not successful due to the lack of surface expression of the channel and the majority of the expressed proteins being localized in the endoplasmic reticulum (ER) and nuclear membranes. A series of mutant constructs with N-terminal deletion was generated and expressed in HEK cells. One of the constructs, LjCASTOR-2TM that contains the pore-forming TM domains (TM3 and 4) and the entire soluble domain, yielded a small amount of plasma membrane-targeted channels, allowing us to measure the channel activity by single channel recording.

Figure 1a shows the whole-cell current in the presence of symmetrical 150 mM $Na^+$, illustrating the cationic ($Na^+$) currents conducted by LjCASTOR-2TM. The channel has a high single-channel conductance of 220 pS in symmetrical 150 mM $Na^+$. In the presence of cytosolic $Mg^{2+}$ (i.e., 1 mM), the channel conduction is slightly inwardly rectified (Fig. 1b). Furthermore, the recordings with inside-out patches revealed that the channel gating is dependent on the cytosolic (bath solution) calcium concentration (Fig. 1c). At a lower concentration range ($\leq 100\,\mu M$), an increase of $[Ca^{2+}]$ in the bath solution resulted in an increase of the channel open probability; at higher concentration range ($\geq 100\,\mu M$), however, $Ca^{2+}$ exerts an inhibitory effect on the open probability, resulting in a bell-shaped $Ca^{2+}$ activation curve reminiscent of $Ca^{2+}$ activation in RYR or IP3 receptors[26–28].

Although CASTOR was initially suggested to function as a $K^+$ channel[12,24], it lacks the highly conserved TVGYG filter sequence of canonical $K^+$ channels, and its selectivity property remains unclear. We, therefore, measured the relative selectivity of LjCASTOR-2TM among $Na^+$, $K^+$, and $Ca^{2+}$ by single-channel recording in the outside-out patch. As shown in Fig. 1d, replacing 150 mM $Na^+$ with 150 mM $K^+$ in the bath solution yielded a reversal potential of about $-20$ mV, indicating that LjCASTOR-2TM is slightly selective for $Na^+$ over $K^+$ with a permeation ratio, $P_{Na}/P_K$, of about 2. Replacing bath $Na^+$ with 15 mM $Ca^{2+}$ yielded a reversal potential of about 30 mV, indicating higher $Ca^{2+}$ selectivity of LjCASTOR-2TM with a permeation ratio, $P_{Ca}/P_{Na}$,

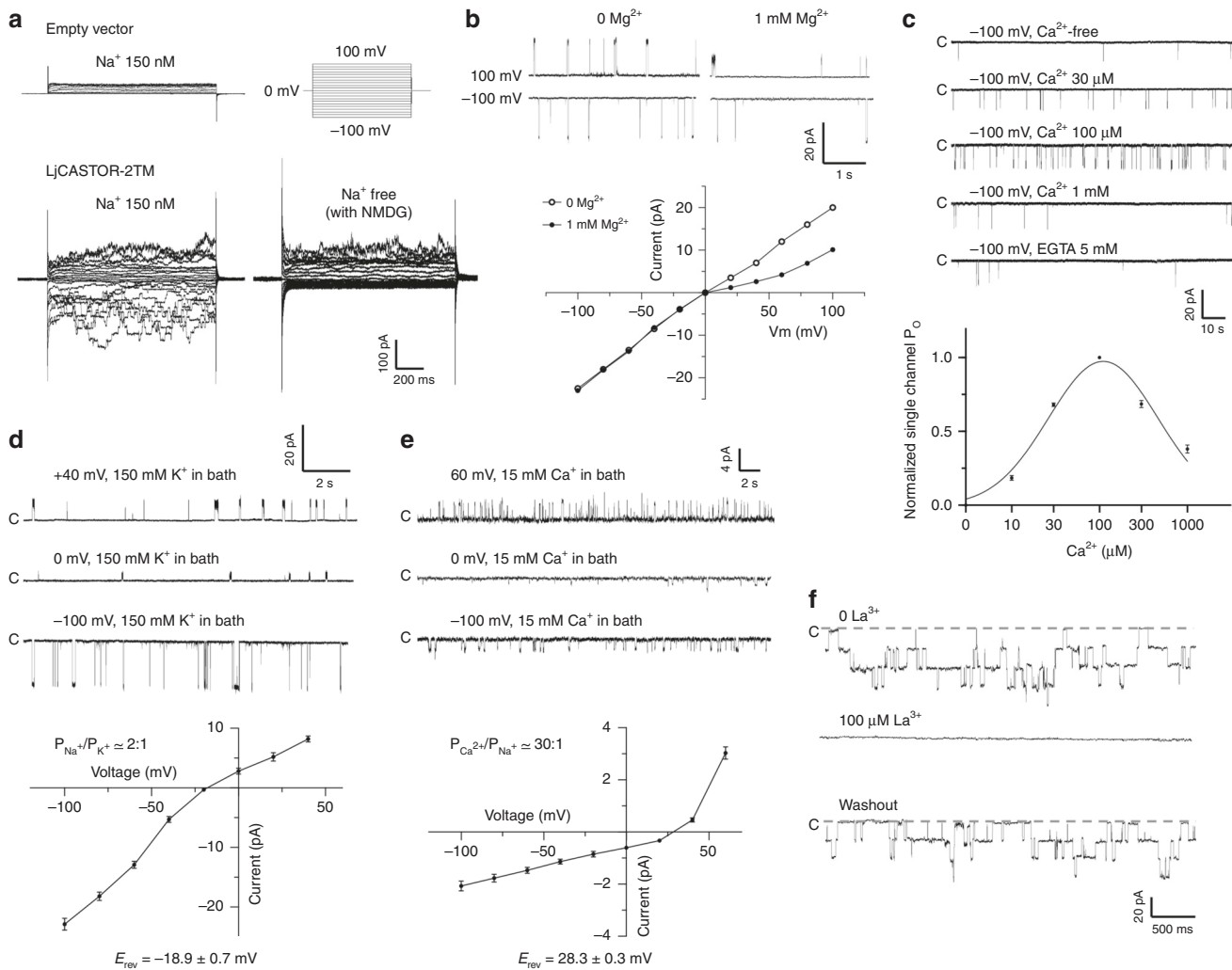

**Fig. 1** Functional characterization of LjCASTOR-2TM reveals that LjCASTOR is a Ca$^{2+}$-regulated Ca$^{2+}$ channel. **a** Whole-cell current recordings in the presence of 150 mM [Na$^+$] and 10 μM [Ca$^{2+}$] in the pipette solution (intracellular) and 150 mM [Na$^+$] or NMDG in the bath solution. LjCASTOR-2TM is not permeable to NMDG. **b** Inward rectification of the current caused by intracellular Mg$^{2+}$ ions. The recordings were obtained from the inside-out patches with the symmetrical 150 mM [Na$^+$] in the presence of 10 μM [Ca$^{2+}$] and 0 or 1 mM [Mg$^{2+}$] in the bath solution (intracellular). **c** Cytosolic [Ca$^{2+}$]-dependent activation of LjCASTOR-2TM. The recordings were obtained from the inside-out patches with 150 mM [Na$^+$] in the pipette solution (extracellular). The bath solution (intracellular) contained 150 mM [K$^+$] with various [Ca$^{2+}$]. Data points are mean ± s.e.m. ($n = 5$ independent experiments). **d** Selectivity of LjCASTOR-2TM between Na$^+$ and K$^+$. Sample traces and I–V curve were obtained from the outside-out patches with 150 mM [Na$^+$] and 10 μM [Ca$^{2+}$] in the pipette solution (intracellular) and 150 mM [K$^+$] in the bath solution (extracellular). P$_{Na^+}$/P$_{K^+}$ value was calculated from the reversal potential ($E_{rev}$ = mean ± s.d. of five independent experiments). **e** Selectivity of LjCASTOR-2TM between Na$^+$ and Ca$^{2+}$. Sample traces and I–V curve were obtained from the outside-out patches with 150 mM [Na$^+$] and 10 μM [Ca$^{2+}$] in the pipette solution (intracellular) and 15 mM [Ca$^{2+}$] in the bath solution (extracellular). P$_{Ca^{2+}}$/P$_{Na^+}$ value was calculated from the reversal potential ($E_{rev}$ = mean ± s.d. of five independent experiments). **f** Inhibition of LjCASTOR-2TM by La$^{3+}$. The recordings were obtained from the outside-out patches with 150 mM [Na$^+$] and 10 μM [Ca$^{2+}$] in the pipette solution (intracellular) and 150 mM [Na$^+$] and 0 or 100 μM [La$^{3+}$] in the bath solution (extracellular)

of about 30. Under our recording condition, LjCASTOR-2TM has a Ca$^{2+}$ conductance of ~11 pS. Furthermore, La$^{3+}$, a lanthanide inorganic blocker of some Ca$^{2+}$-permeating channels, can also block LjCASTOR-2TM from the extracellular side (Fig. 1f). Taken together, LjCASTOR-2TM functions as a Ca$^{2+}$-regulated Ca$^{2+}$ channel.

It is worth noting that LjCASTOR-2TM exhibits much higher single-channel activity in the cell-attached configuration with low endogenous [Ca$^{2+}$] than that measured in the inside-out patch (Supplementary Fig. 1). Recording of whole-cell current also showed a time-dependent decrease in channel activity after break-in. This observation raises the possibility that other unknown cellular factors could also play important roles in channel gating. Further study is needed to address this possibility.

**The CASTOR soluble domain forms RCK gating ring**. The soluble region of CASTOR constitutes two-thirds of the protein and likely functions as the ligand-binding domain of the channel. To reveal the structural basis of ligand gating in CASTOR, we expressed and purified the post-TM soluble region of the protein and determined its structure by X-ray crystallography. The protein crystals had a space group of $I4$ and contained one subunit per asymmetric unit; four crystallographic symmetry-related subunits assembled into a tetramer with the molecular fourfold coinciding with the crystallographic tetrad. The final structure was refined to 1.6 Å with an $R_{work}$ of 16.59% and an $R_{free}$ of 18.82% (Methods and Supplementary Table 1).

Despite the lack of sequence homology, the overall structure of the CASTOR soluble region is similar to that of the RCK gating

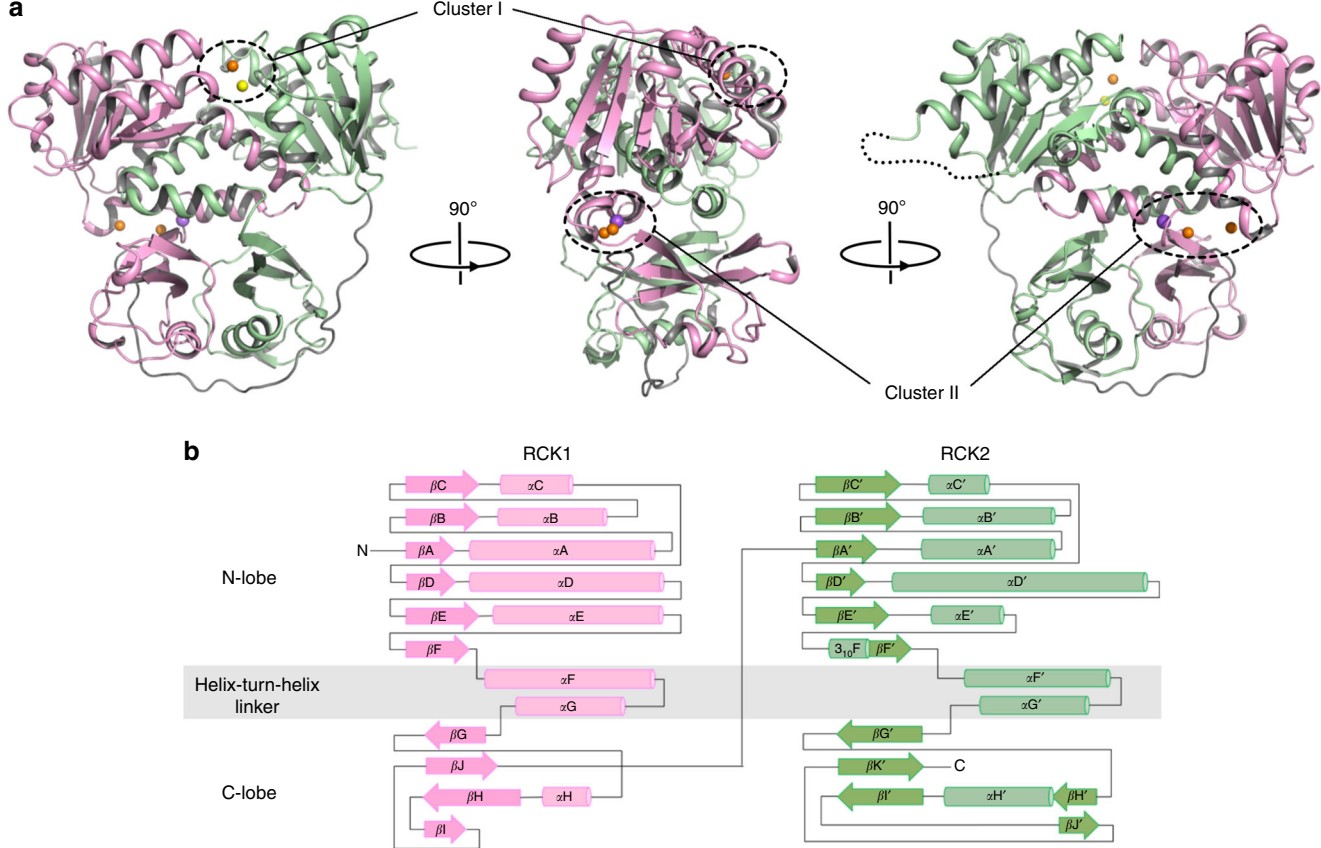

**Fig. 2** Structure of a single LjCASTOR soluble subunit. **a** The LjCASTOR soluble subunit contains two RCK domains with two clusters of ion-binding sites: pink, RCK1 (residues 320–552); green, RCK2 (residues 581–853); gray, RCK1–RCK2 linker (553–580); a dotted line, residues 699–726 with no electron density; orange sphere, $Ca^{2+}$; yellow sphere, $Mg^{2+}$; purple sphere, $Na^+$. **b** Topology and domain organization of the two RCK domains of LjCASTOR

ring observed in ligand-gated $K^+$ channels such as human BK channel: each soluble subunit consists of two tandem RCK domains (RCK1: residues 320–552; RCK2: residues 581–853) and four subunits assemble into a tetrameric gating ring, confirming that the CASTOR channel functions as a tetramer (Figs. 2 and 3 and Supplementary Figs. 2 and 3).

Each RCK domain can be divided into three subdomains: an N-terminal Rossmann-folded subdomain termed N-lobe (from βA to βF); a C-terminal subdomain (C-lobe) that interacts with its counterpart from the neighboring RCK domain in the same subunit; and a helix-turn-helix linker between the N- and C-lobes that interlocks the two tandem RCK domains within each subunit (Fig. 2). The tandem RCK domains form a bi-lobed architecture with extensive inter-RCK interactions at the helix-turn-helix and C-terminal subdomain. The two RCK domains are covalently connected by a long but well-structured linker that wraps around the C-terminal subdomains and ties them to the N-lobe of RCK2 like a rope (Fig. 2 and Supplementary Fig. 2).

Four CASTOR soluble subunits assemble into a tetrameric gating ring consisting of eight RCK domains (Fig. 3a). The Rossmann-folded N-lobes constitute the central core of the gating ring while the C-terminal subdomains form four protruded knobs at its periphery (Fig. 3a). Three sets of inter-subunit interactions define the gating-ring assembly (Fig. 3b, c). The most significant interactions occur at the interface formed by helices αD and αE from RCK1 and the equivalent helices from RCK2 of the neighboring subunit. This interface was termed assembly interface in RCK-regulated $K^+$ channels[15]. The protein–protein contacts at the assembly interface are mainly hydrophobic along with a salt bridge between Arg 420 and Asp 735. In addition to

the protein contact at the assembly interface, two sets of inter-subunit salt-bridge interactions are noticeable: one is the Lys368–Asp390 salt bridge between two neighboring RCK1s at the top of the gating ring near the membrane surface, and the other set of interactions are between residues from two neighboring RCK2s at the bottom of the gating ring, including His651–Asp626 and Arg650–Asp 626.

**Ion-binding sites in the gating ring**. The CASTOR soluble domain was crystallized in the presence of $Na^+$, $Ca^{2+}$, and $Mg^{2+}$, and its high-resolution structure allowed us to identify multiple bound ions clustered in two locations within each subunit. The identity of the bound ions was determined by the chemical environment of the ions such as ligand type, coordination numbers, and ion–ligand distances, along with the anomalous scattering signal from $Ca^{2+}$, which is much stronger than that of $Na^+$ or $Mg^{2+}$.

Cluster I contains two bound ions and is located in the cleft between the two N-lobes (Fig. 2a, 4a). One ion is defined as $Mg^{2+}$; it is coordinated by five water molecules and one carboxylate oxygen from D444 with an averaged O–Mg distance of about 2.1 Å, reminiscent of octahedral $[Mg(H_2O)_6]^{2+}$. The other ion is defined as $Ca^{2+}$ and is labeled as the Ca(I) site; it is coordinated by eight oxygen atoms including multiple negatively charged carboxylates, and it has ion–ligand distances of 2.4–2.5 Å (Fig. 4a), both of which are commonly seen for $Ca^{2+}$ binding in proteins. The $Ca^{2+}$ identity was further confirmed by its anomalous scattering signal in the anomalous difference Fourier map, which is not observed at the $Mg^{2+}$ site (Supplementary Fig. 4). The $Mg^{2+}$ binding appears to contribute to the observed

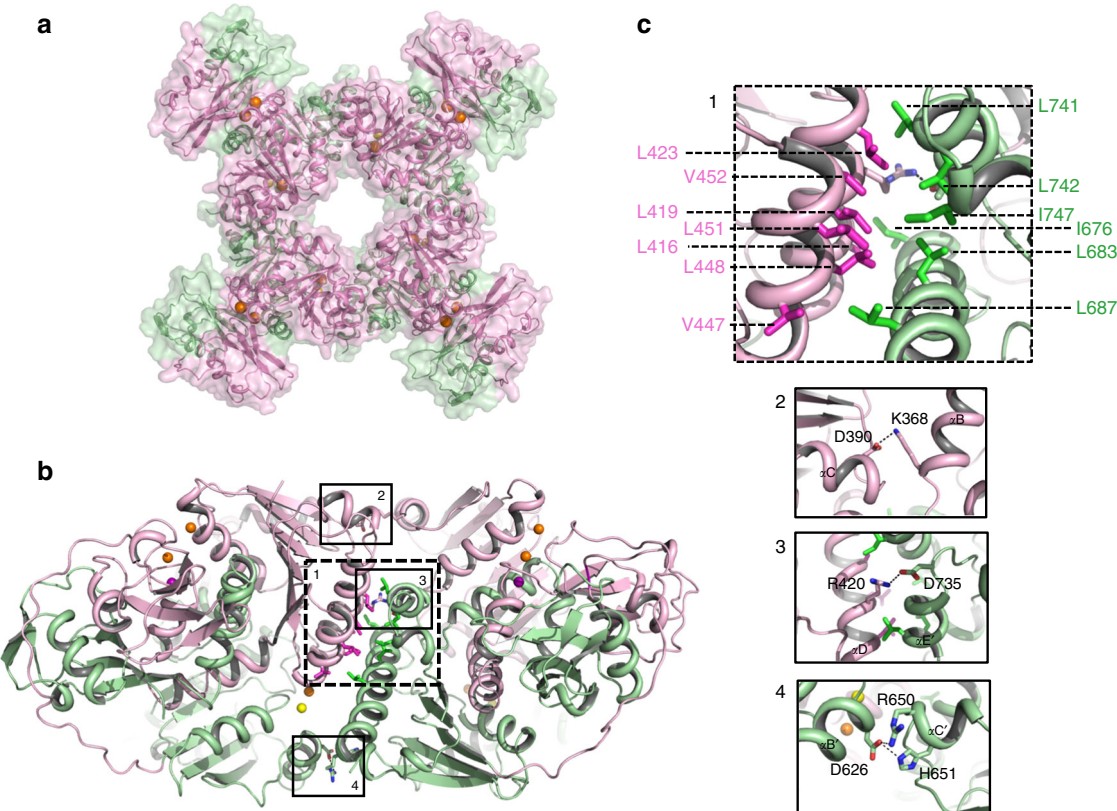

**Fig. 3** Structure of the $Ca^{2+}$-bound LjCASTOR gating ring composed of four subunits. **a** Top view of the crystal structure of the LjCASTOR gating ring with the same color representation as in Fig. 2a. **b** Side view of LjCASTOR gating ring and inter-subunit interactions. **c** Zoomed-in views of the boxed regions in **b**. The dashed box (c1) highlights the residues in the assembly interface involved in the network of hydrophobic interactions between αD and αE in RCK1 and αD′ and αE′ in RCK2 from the neighboring subunit. The main salt bridges on the inter-subunit interface are illustrated in the solid boxes (c2–4)

inward rectification of the channel conductance as D444A mutant, the single amino acid that directly chelates $Mg^{2+}$, abolished the $Mg^{2+}$-dependent rectification of the channel (Supplementary Fig. 5). Cytosolic $Mg^{2+}$ is known to block the pore and cause inwardly rectified channel conduction in Kir channels[29]. However, it is unclear how $Mg^{2+}$ modulates the outward ion conduction in CASTOR, as the $Mg^{2+}$ is too far from the ion conduction pore to be a blocker. The $Ca^{2+}$ binding at the Ca(I) site mediates extensive interactions between the two N-lobes that lock them in a fixed position relative to each other. These interactions consist of a network of hydrogen bonds and salt bridges (Fig. 4a). Notably, the side chain of His 463 inserts into the gap between Arg 589 and 590, and forms π-cation planar stacking interactions with their guanidinium groups. One of the $Ca^{2+}$-chelating residues, D442, directly participates in the inter-lobe contact by forming a salt bridge with R590 (Fig. 4a and Supplementary Fig. 6).

Cluster II sites are located in the cleft between the N- and C-terminal subdomains of RCK1 and contain three bound ions. The anomalous signals indicate that only two of the bound ions are $Ca^{2+}$, labeled as Ca(II) and Ca(III) sites, respectively (Fig. 4b and Supplementary Fig. 4b). The Ca(II) site is at the inner part of the cleft, and the ion is chelated by seven oxygen ligands (Fig. 4b). Only one acidic residue, E493, participates in bidentate ion chelation with its side-chain carboxylate. The rest of the ligands are from two water molecules and the backbone carbonyls of F489, N491, and E552. The Ca(III) site is at the outer part of the cleft and the $Ca^{2+}$ is also coordinated by seven oxygen ligands, including three water molecules, the backbone carbonyl of A342, side chain hydroxyl of S345 and side chain carboxylates of D553 and D554. Both sites have ion–ligand distances in the range of

2.4–2.5 Å (Fig. 4b), optimal for $Ca^{2+}$. The location of the two $Ca^{2+}$ sites indicates that $Ca^{2+}$ binding provides two tethering points between the N-lobe of RCK1 and the periphery C-terminal subdomains. The third ion in cluster II is in close proximity to the Ca(II) site and was identified as $Na^+$ for two reasons: first, the bound ion has no anomalous diffraction signal; second, the bound ion has a coordination number of five with ion–ligand distances of 2.3–2.4 Å, which match that of the $Na^+$ ion. Other than one carboxylate oxygen atom from E493, which is also shared with the Ca(II) site, the rest of the $Na^+$ ligands are all from backbone carbonyls of D485, G488, E490, and C492 (Fig. 4b). The $Na^+$ site is not $Na^+$ specific as the structure determined by using crystals grown in the presence of KCl instead of NaCl demonstrated that $K^+$ could also bind (Fig. 5a, b).

Interestingly, $K^+$ binding triggers some local structural changes, most notably the flip of the backbone carbonyl of L487, resulting in two additional oxygen ligands for $K^+$ from the backbone carbonyls of I486 and L487. The $K^+$ position is also slightly shifted from the $Na^+$ site, resulting in longer ion–ligand distances in the range of 2.6–3.0 Å (Fig. 5a, b). It is intriguing to note that $Na^+$ and $K^+$ can modulate the open probability of the channel. As shown in Fig. 5c, with the same cytosolic $[Ca^{2+}]$, the channel has higher open probability in KCl than in NaCl. As this monovalent site is proximal to the Ca(II) site and also shares the same oxygen ligand from E493, we suspect that $Na^+$ and $K^+$ binding can exert a different competing effect on $Ca^{2+}$ affinity and thereby have different modulation effect on $Ca^{2+}$ activation.

**Apo structure of LjCASTOR gating ring.** In RCK-regulated $K^+$ channels such as BK and MthK, the gating rings undergo

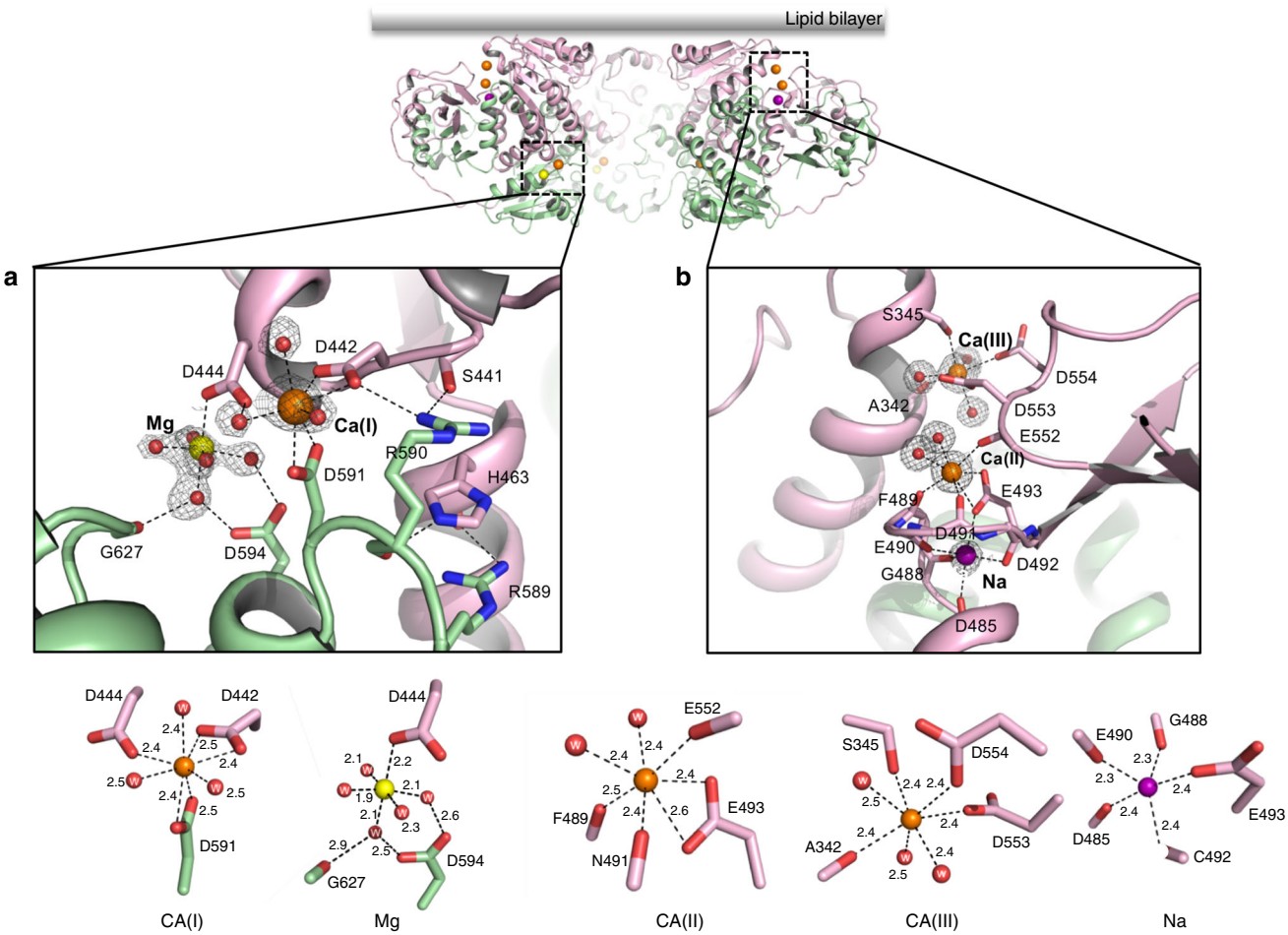

**Fig. 4** The locations and structure details of the two clusters of ion-binding sites within each LjCASTOR subunit. **a** Structure of the cluster I ion-binding sites (top panel) with detailed ion coordination distances (bottom panel). **b** Structure of the cluster II ion-binding sites (top panel) with detailed ion coordination distances (bottom panel). The electron density (gray mesh contoured at 1.5 σ) of ions and water molecule is calculated from the 2Fo-Fc map

conformational changes in response to $Ca^{2+}$ binding and open the pore[30–32]. To reveal if $Ca^{2+}$ binding induces a similar conformational change in CASTOR, we also crystallized the apo form of CASTOR gating ring with the presence of 1 mM ethylene glycol-bis(β-aminoethyl ether)-N,N,N′,N′-tetraacetic acid (EGTA). Compared to the $Ca^{2+}$-bound form, CASTOR gating ring is biochemically less stable in the absence of $Ca^{2+}$ as demonstrated by the broader and less symmetrical protein elution peak on a gel-filtration column. The obtained crystals diffracted to a much lower resolution and the final structure was determined to 3.3 Å.

The crystal has a space group of P $2_1$ and contains four subunits per asymmetric unit; the four subunits exhibit slight conformational heterogeneity among them and form an asymmetrical gating ring (Fig. 6a, b and Supplementary Fig. 7). In comparison to the $Ca^{2+}$-bound state, the structural change between apo and $Ca^{2+}$-bound subunits mainly involves rigid body movement of the N-lobe of RCK1, causing alterations in relative orientations between the subdomains (Fig. 6c). In the $Ca^{2+}$-bound state, the N-lobe of RCK1 is tightened to the rest of the protein via two $Ca^{2+}$-mediated interactions at cluster I and cluster II sites, respectively. Both interactions are missing in the apo state and consequently the RCK1 N-lobe has more freedom to undergo rigid body movement: at cluster I site, the RCK1 N-lobe slides apart from the RCK2 N-lobe; at the cluster II site, the RCK1 N-lobe swings away from the C-terminal subdomain, resulting in a slightly enlarged cleft between the two. In the

context of gating ring, however, there is no significant change in the ring size of CASTOR with or without $Ca^{2+}$. As shown in Fig. 6a, the two diagonal distances of the asymmetrical apo ring (between the Cα of Thr 323) are about 61 and 66 Å, respectively, whereas the equivalent distance is 62 Å in the symmetrical $Ca^{2+}$-bound gating ring (Fig. 6b). This finding is different from BK or MthK channels in which $Ca^{2+}$ ligand induces a significant expansion of the gating ring. In light of the low single-channel, open probability of CASTOR with $Ca^{2+}$ activation as well as the inhibition of channel by $Ca^{2+}$ at sub-millimolar concentrations, the structures of the CASTOR gating rings in the apo and $Ca^{2+}$-bound states both likely represent the closed conformation.

**Legume nodule development requires gating ring $Ca^{2+}$ binding.** A main biological function of these nuclear ion channels in legumes is the generation of $Ca^{2+}$ spiking in response to rhizobia, which, in turn, regulates symbiotic gene expression and allows the development of nitrogen-fixing root nodules and rhizobial infection. We, therefore, investigated if the $Ca^{2+}$ binding in the CASTOR gating ring is essential for the establishment of root nodule symbiosis between *L. japonicus* and its compatible rhizobia (*Mesorhizobium loti*). In this assay, *Ljcastor-4* mutant plants were transformed with empty vector (negative control), a vector expressing the wild-type CASTOR protein (positive control), or CASTOR mutants with point mutations at the $Ca^{2+}$-binding sites. After plants established robust root systems, they were transplanted into pots, inoculated with the compatible rhizobium

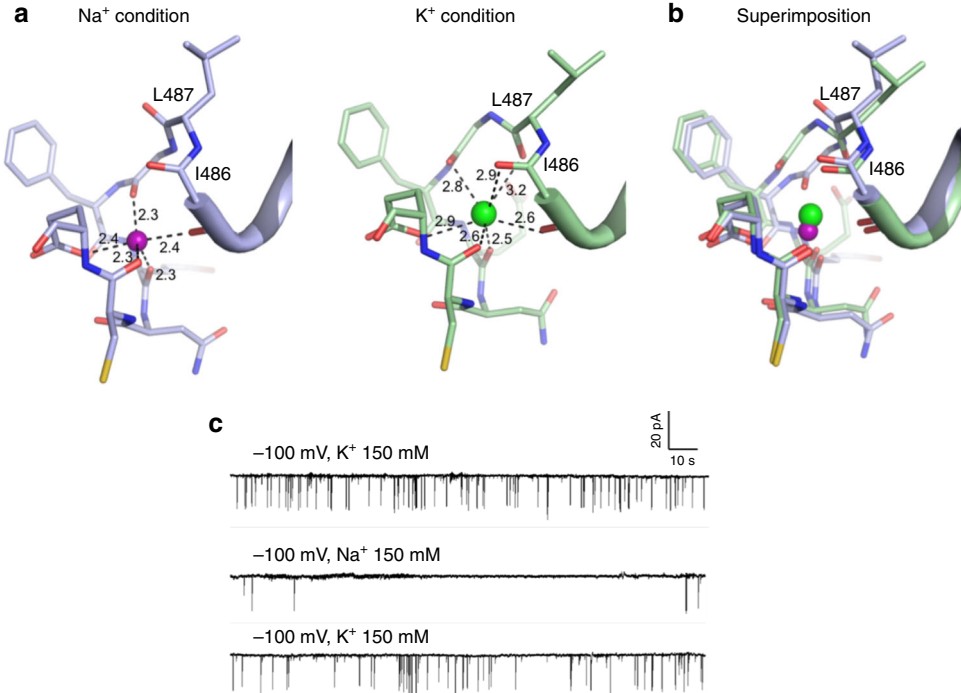

**Fig. 5** K$^+$ binding at the cluster II Na site. **a** Structural comparison of the cluster II Na site between Na$^+$-bound (left) and K$^+$-bound (right) structures. Purple and green spheres represent Na$^+$ and K$^+$ ions, respectively. **b** Superimposition of the two structures. **c** Single channel recordings from an inside-out patch with 150 mM [Na$^+$] in the pipette solution (extracellular) and 150 mM [Na$^+$] or [K$^+$] in the bath solution (intracellular). Replacing K$^+$ with Na$^+$ in the bath solution significantly reduced the single-channel open probability

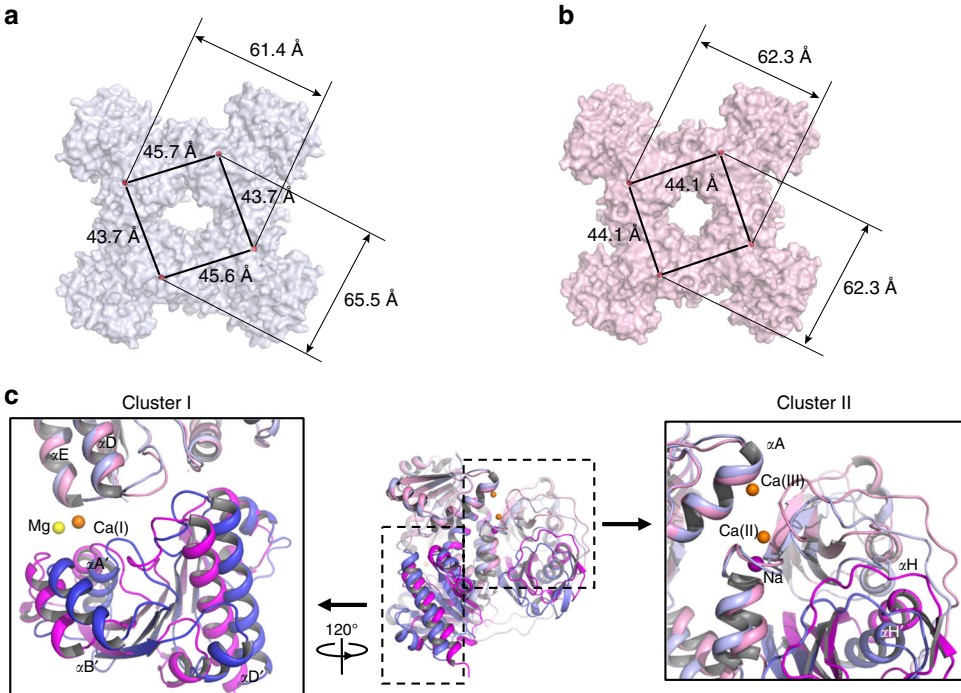

**Fig. 6** Crystal structure of the apo LjCASTOR gating ring. **a** Surface-rendered representation of the apo gating ring, which is no longer fourfold symmetrical. Distances are measured between the C$\alpha$ atoms of Thr 323 s from two neighboring or diagonal subunits. **b** Surface-rendered representation of the Ca$^{2+}$-bound gating ring. **c** Single subunit superimposition between Ca$^{2+}$-bound (RCK1 in pink and RCK2 in magenta) and apo (RCK1 in light blue and RCK2 in blue) structures. The RCK1 N-lobes were used in the structural alignment. Insets are zoomed-in view of the structural differences at the cluster I (left) and II sites (right). In each inset, the other subdomains of no interest were removed for visual clarity

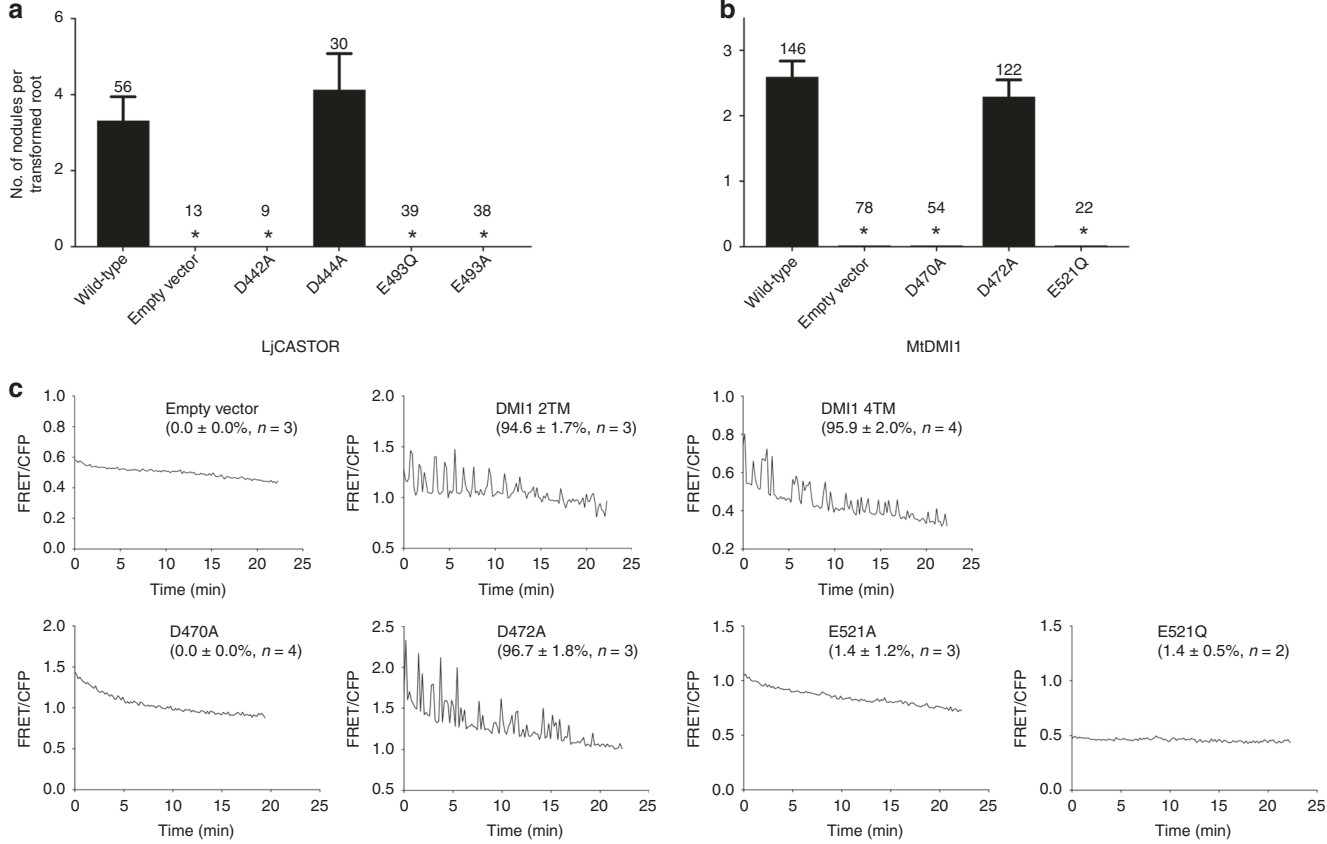

**Fig. 7** Functional importance of gating ring $Ca^{2+}$ binding. **a** Nodulation assay for LjCASTOR and its $Ca^{2+}$ site mutants expressed in CASTOR-null *L. japonicus*; the number above the error bar indicates the number of roots tested; *, $p < 3.1 \times 10^{-4}$. **b** Nodulation assay for MtDMI1 and its $Ca^{2+}$ site mutants expressed in DMI1-null *M. truncatula*; *, $p < 2.6 \times 10^{-6}$. **c** Representative recordings of calcium spiking in HEK 293 cells transfected with full-length MtDMI1 and its variants or MtDMI1-2TM. The numbers in the parenthesis are the population ratios of the cells that were positive in $Ca^{2+}$ spiking (mean ± s.e.m. in percentage and $n$ is the number of measurements)

*M. loti* and grown under 16-h light/8-h dark conditions at 23 °C for 3 weeks before being harvested to count the nodules. The efficiency of symbiosis formation was determined by calculating the average number of nodules per transformed root. Mutations at two of the $Ca^{2+}$-binding sites were tested: D442A and D444A at the cluster I Ca(I) site and E493A or Q at cluster II Ca(II) site. As shown in Fig. 7a, plants expressing D442A, E493A, or E493Q mutants failed to form root nodules upon rhizobia infection demonstrating the importance of $Ca^{2+}$ binding for the physiological function of CASTOR. Interestingly, mutation of D444, which also participates in Ca(I) coordination, did not affect the symbiosis.

As all $Ca^{2+}$ sites are highly conserved in these nuclear channels (Supplementary Fig. 2), we also mutated the corresponding $Ca^{2+}$ site residues in DMI1 and performed a similar nodulation rescue assay on the *M. truncatula dmi1-4* (FN1) mutant with *Sinorhizobium meliloti*. The result was consistent with that observed in the CASTOR mutations: plants with D470A (equivalent to D442A of CASTOR) and E521Q (equivalent to E493Q of CASTOR) mutations failed to form root nodules, whereas plants with D472A (equivalent to D444A of CASTOR) mutation established normal root nodules similar to those with wild-type channel (Fig. 7b).

**Gating ring $Ca^{2+}$ binding is essential for $Ca^{2+}$ spiking.** To test if the loss of root nodulation in those $Ca^{2+}$ site mutations is correlated to the loss of the channel's ability to generate $Ca^{2+}$

spiking, we took advantage of the FRET-based calcium oscillation assay in DMI1-expressing HEK 293 cells. This assay is based on the observation that expressing DMI1 in HEK 293 cell allows $Ca^{2+}$-induced $Ca^{2+}$ spiking in the cytosol, recapitulating the plant nuclear $Ca^{2+}$ spiking upon DMI1 activation. The reason to use DMI1 instead of CASTOR in this assay is that the $Ca^{2+}$-induced $Ca^{2+}$ oscillation is not observed in CASTOR-expressing HEK cells. This could be because DMI1 alone is sufficient to initiate $Ca^{2+}$ spiking in *M. truncatula* whereas both POLLUX and CASTOR are required in *L. japonicus*[12].

In this assay, DMI1 was co-expressed with Yellow Cameleon 3.6 (YC3.6) in HEK 293 cells. YC3.6 consists of CFP (cyan) and YFP (yellow) fluorophores and yields a $[Ca^{2+}]$-dependent FRET that was used for monitoring cytosolic $Ca^{2+}$ concentration change[33]. The cytosolic $Ca^{2+}$ spiking in DMI1-expressing cells was induced by stepwise increase of extracellular $[Ca^{2+}]$. As shown in Fig. 7c, the addition of $Ca^{2+}$ in bath solution to 4 mM induces $Ca^{2+}$ spiking in DMI1-expressing cells as demonstrated by the oscillations of FRET signals between CFP and YFP (Supplementary Movie 1). This oscillation is not observed in control cells without expressing DMI1. Using the 2TM construct of DMI1 equivalent to CASTOR-2TM yielded similar $Ca^{2+}$-induced $Ca^{2+}$ spiking similar to the wild-type channel, confirming that the removal of the first two TM helices does not affect the channel function. Consistent with root nodulation assay, cells expressing DMI1 mutants (D470A, E521A, and E521Q) that abolish root nodulation in *M. truncatula* also failed to trigger

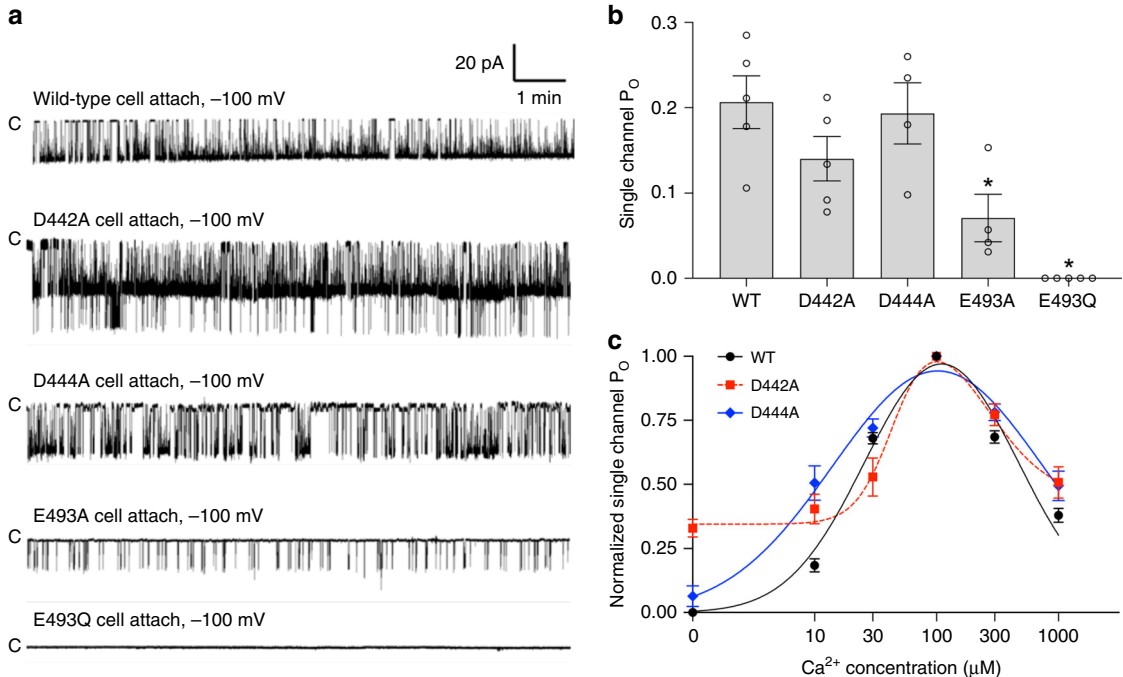

**Fig. 8** Single-channel activity of the $Ca^{2+}$ site mutants of LjCASTOR-2TM. **a** Sample traces of single-channel recordings of the $Ca^{2+}$ site mutants from cell-attached patches. **b** Single-channel open probability of the $Ca^{2+}$ site mutants of LjCASTOR-2TM; the number above the error bar indicates the number of measurements; *, $p < 0.05$. **c** Effect of Ca(I) site mutations on the cytosolic $[Ca^{2+}]$-dependent activation of LjCASTOR-2TM. The recordings were obtained from the inside-out patches with 150 mM $[Na^+]$ in the pipette solution (extracellular). The bath solution (intracellular) contained 150 mM $[K^+]$ with various $[Ca^{2+}]$. Data points are mean ± s.e.m. ($n = 5$ independent experiments)

$Ca^{2+}$ spiking, whereas the D472A mutant that does not affect nodulation yielded similar $Ca^{2+}$ spiking as the wild-type channel.

**Single-channel activities of CASTOR $Ca^{2+}$ site mutations**. It was intriguing that D442A and D444A mutants at the same $Ca^{2+}$ site have the opposite effect on root nodulation and $Ca^{2+}$ spiking. To correlate the effect of $Ca^{2+}$ sites mutations on root nodulation and $Ca^{2+}$ oscillation with the electrophysiological property of channel, we measured the single-channel activity of these $Ca^{2+}$ site mutations by patch clamp in cell-attached configuration (Fig. 8a, b). As expected, E493A mutant has much lower single-channel activity than wild-type channel, whereas no channel activity was observed for E493Q mutant, indicating that the loss of nodulation and $Ca^{2+}$ spiking in Ca(II) site mutations can be attributed to the loss of channel function. On the contrary, both D442A and D444A mutants exhibit similar single-channel Po as the wild-type channel. However, these two mutants showed different $[Ca^{2+}]$-dependent activation in inside-out patches. While the D444A mutant exhibits similar bell-shaped $Ca^{2+}$ activation as the wild-type channel, D442A has higher basal channel activity at low $Ca^{2+}$, resulting in a leaky channel even in the absence of $Ca^{2+}$ (Fig. 8c). It is worth noting that there is one key structural difference between D442 and D444. In addition to ion chelation at the Ca(I) site, D442, but not D444, also directly participates in the inter-RCK interactions at the $Ca^{2+}$-bound state that lock the two N-lobes of RCK domains (Fig. 4a and Supplementary Fig. 6). This key difference could potentially contribute to the functional difference between the D442A and D444A mutants.

## Discussion

Using LjCASTOR as our model system, we demonstrated that the plant nuclear ion channels, including DMI1, POLLUX, and CASTOR, function as tetrameric $Ca^{2+}$-regulated $Ca^{2+}$ channels whose ligand-binding domains form an RCK gating ring. Distinct

from RCK-containing $K^+$ channels, these plant channels lack the TVGYG signature sequence essential for $K^+$ selectivity. We demonstrated that CASTOR exhibits high $Ca^{2+}$ selectivity over $K^+$ or $Na^+$. This finding suggests that DMI1, POLLUX, and CASTOR can by themselves function as $Ca^{2+}$ release channels in the nucleus and thereby negate the previous suggestion that they function as $K^+$ channels. CASTOR has a predicted filter sequence of $AD^{265}S(A)GNHA$ that is highly conserved among these plant nuclear channels. We suspect D265 plays an essential role in $Ca^{2+}$ selectivity. Neutralization of this acidic residue by D265N mutation completely abolished the channel activity as the mutant failed to elicit any current in the whole-cell recordings (Supplementary Fig. 8), suggesting that this filter aspartate is critical for channel conduction. Structural determination of the intact channel with the ion conduction pore will be needed to reveal the structural basis of $Ca^{2+}$ selectivity of these channels.

$Ca^{2+}$ regulates CASTOR gating in a biphasic manner similar to that observed in IP3 and ryanodine receptors: the channel open probability is potentiated at low $Ca^{2+}$ concentrations but inhibited at higher $Ca^{2+}$ concentrations, resulting in a bell-shaped $Ca^{2+}$ activation curve. This property leads us to suggest that CASTOR, POLLUX, and DMI1 have a similar $Ca^{2+}$-induced $Ca^{2+}$ release (CICR) function to IP3 and ryanodine receptors, two classical $Ca^{2+}$ release channels in ER and sarcoplasmic reticulum (SR), respectively. Our high-resolution structure reveals multiple $Ca^{2+}$-binding sites in the RCK gating ring and provides structural insights into $Ca^{2+}$-regulated gating of CASTOR. It is worth noting that while the RCK-regulated MthK and BK channels are also $Ca^{2+}$-gated, their $Ca^{2+}$-binding sites within the RCK gating ring are completely different from those in CASTOR (Supplementary Fig. 3).

In RCK-regulated $K^+$ channels, the ligand binding can expand the gating ring and open the pore. In the example of BK channels, the ligand-induced gating-ring expansion is caused by the rigid body movement of the N-lobe of RCK1[30,32]. As the N-lobe of

RCK1 is directly linked to the ion conduction pore of the channel, its movement likely determines the opening and closing of the channel. Similarly, the N-lobe of CASTOR RCK1 appears to be the most mobile part of gating ring when comparing the structures of $Ca^{2+}$ bound and apo states, and we suspect its movement could trigger similar gating-ring conformational change as BK channels. However, the biphasic $Ca^{2+}$ regulation of CASTOR implies that $Ca^{2+}$ binding in its gating ring can either promote or inhibit the channel opening. The strategic positions of the two clusters of ion-binding sites lead us to speculate that $Ca^{2+}$ binding at the cluster I Ca(I) site plays an inhibitory role whereas $Ca^{2+}$ binding at the cluster II Ca(II) and Ca(III) sites activates the channel (Supplementary Fig. 9). That is because $Ca^{2+}$ at the Ca(I) site mediates inter-lobe interactions between the two N-lobes within each subunit that would prevent the RCK1 N-lobe from swinging away from the central axis and expanding the ring. Cluster II sites, on the other hand, are located at the cleft between RCK1 N-lobe and the peripheral C-terminal subdomains; $Ca^{2+}$ binding tethers the two parts of the protein by moving RCK1 N-lobe away from the center and thereby promotes gating-ring expansion.

With $Ca^{2+}$ as the sole ligand, CASTOR has a low single-channel activity when it is isolated from the cellular environment as demonstrated in the recording of inside-out patches. On the contrary, in the cell-attached configuration, the channel exhibits much higher activity with low endogenous $Ca^{2+}$ (Supplementary Fig. 1). This observation suggests that some other proteins or unidentified cellular factors in addition to $Ca^{2+}$ could also participate in channel activation. This situation is somewhat similar to IP3 and ryanodine receptors whose functions are regulated by multiple ligands[34–36]. We showed recently that metabolites of the mevalonate (MVA) pathway are involved in the symbiotic signaling pathway in *M. truncatula*[13]. It was demonstrated that MVA could elicit nuclear $Ca^{2+}$ oscillation. Further studies will be needed to demonstrate whether MVA or its derivatives could directly bind and regulate these symbiotic channels, or whether other unidentified secondary messengers activate the channels.

Intriguingly, overexpression of DMI1 in human HEK 293 cells can allow $Ca^{2+}$-induced cytosolic $Ca^{2+}$ spiking. A similar phenomenon has also been observed when the ryanodine receptor was expressed in the same type of cells[37,38]. As the overexpressed DMI1 is predominantly localized in ER and nuclear membranes (Supplementary Fig. 10), it likely participates in the $Ca^{2+}$ spiking by functioning as an ER $Ca^{2+}$ release channel in lieu of IP3 receptors. In the native environment, the DMI1 family channels are localized in the membrane of the nuclear envelope that is continuous with the ER. We suspect they play a similar functional role as a $Ca^{2+}$ release channel in generating $Ca^{2+}$ spiking. A recent study demonstrated that nuclear-localized, $Ca^{2+}$-permeable CNGC channels, CNGC15s, can directly interact with DMI1 and mediate calcium spiking[25]. Further study is needed to fully understand the interplay between these two groups of nuclear channels in generating the $Ca^{2+}$ spiking, and also to identify other critical components in the feedback loop that triggers $Ca^{2+}$ spiking.

## Methods

**Cloning, expression, and purification of LjCASTOR RCK domain**. A PCR fragment encoding the soluble domain (residues 312–853) of LjCASTOR was cloned into the pQE60 expression vector containing C-terminal hexahistidine tag. After transformation of *Escherichia coli* SG13009 with the resulting plasmid, the cells were cultured in Luria–Bertani broth medium initially at 37 °C to an optical density ($OD_{600}$) of 0.4, and then at a lowered temperature of 20 °C to $OD_{600}$ of 0.8, when protein overexpression was induced with 0.4 mM isopropyl β-D-thiogalactoside. Cells were harvested 6 h after induction. Pelleted cells were washed with ice-cold lysis buffer base containing 50 mM Tris-HCl, pH 8.0, and 250 mM NaCl, and then resuspended in the lysis buffer supplemented with 4 mM β-

mercaptoethanol (BME), 2 μg mL$^{-1}$ DNase A, and protease inhibitors including 0.1 μg mL$^{-1}$ pepstatin A, 1 μg mL$^{-1}$ leupeptin, 1 μg mL$^{-1}$ aprotinin, and 1 mM phenylmethyl-sulfonyl fluoride (PMSF). Cells were homogenized by sonication with Branson SFX550 Sonifier® (Emerson). After removing the insoluble fraction from the cell lysate by centrifugation at $20,000 \times g$ for 30 min at 4 °C, the supernatant was incubated with a Talon $Co^{2+}$ affinity resin (Clontech) for 2 h on ice with gentle shaking, and the beads were collected in a column with gravity-flow. For the LjCASTOR soluble domain in a calcium-bound state, the beads were first washed with 10 column volumes of buffer A (50 mM Tris-HCl, pH 8.0, 250 mM NaCl, 4 mM BME, 1 mM PMSF, and 1 mM $CaCl_2$), followed by another 10 column volumes of buffer A supplemented with 5 mM imidazole. The protein was then eluted using buffer A supplemented with 300 mM imidazole. The eluate was concentrated using 100 kDa cut-off Amicon Centrifugal Filter (EMD Millipore) and further purified by a Superdex-200 (10/30 GL) size-exclusion column (GE Healthcare) equilibrated in buffer A. Peak fractions were collected and concentrated to 4–5 mg mL$^{-1}$ for crystallization. For protein in the apo state, $CaCl_2$ was excluded from all the buffers used in purification. After $Co^{2+}$ affinity binding, the protein was washed with the washing buffer (50 mM Tris-HCl, pH 8.0, 250 mM NaCl, 4 mM BME, and 1 mM PMSF). The eluate was treated with 5 mM EGTA for 30 min on ice, and any precipitates formed during incubation were removed by centrifugation. The supernatant was concentrated and purified by Superdex-200 column equilibrated in buffer B (50 mM Tris-HCl, pH 8.0, 250 mM NaCl, 4 mM BME, 1 mM PMSF, and 0.5 mM EGTA).

**Crystallization**. Crystallization of the LjCASTOR soluble domain was carried out at 4–5 mg mL$^{-1}$ of protein concentration using the sitting-drop vapor diffusion method. The protein solution was mixed with an equal volume of well solution and incubated at 20 °C. For the protein in the calcium-bound state, crystals appeared within 2 days under the condition of 21% (w/v) PEG 550 MME, 50 mM MgAc$_2$, 100 mM Tris, pH 8.2, and grew to their full size in 2 weeks. Crystals in the sitting drops were cryoprotected by replacing the well solution with a cryosolution containing 40% (w/v) PEG 550 MME, 50 mM MgAc$_2$, 100 mM Tris, pH 8.2, followed by overnight equilibration, and flash-frozen in liquid nitrogen. The protein purified in the presence of EGTA yielded crystals in the solutions containing 20% (w/v) PEG 3350 and 100 mM LiAc. Crystals from the calcium-free condition were first observed within 2 weeks and grew to their full size in 4 weeks. Crystals were cryoprotected by slowly replacing the crystallization drop solution with cryosolution containing 50 mM Tris-HCl, pH 8.0, 250 mM NaCl, 4 mM BME, 1 mM PMSF, 0.5 mM EGTA, 100 mM LiAc, 20% (w/v) PEG 3350, and 20% (w/v) PEG400.

**Data collection and structure determination**. X-ray diffraction data were collected at Advanced Photon Synchrotron (23ID-B, 23ID-D, and 19ID beamlines) and Advanced Light Source (BL 8.2.1 and BL 8.2.2 beamlines). The crystal formed in the presence of $Ca^{2+}$ belongs to space group $I4$ with the unit cell dimensions of $a = b = 125.2$ Å, $c = 81.5$ Å, and $\alpha = \beta = \gamma = 90°$, and contains a single subunit per asymmetric unit; four crystallographic symmetry-related subunits form a gating ring. The protein crystal in the apo state has $P 2_1$ space group with unit cell dimensions of $a = 100.4$ Å, $b = 116.0$ Å, $c = 113.0$ Å, $\alpha = \gamma = 90°$, and $\beta = 113.9°$, and contains four subunits in an asymmetric unit that form a gating ring. Phases were obtained from a single wavelength anomalous dispersion (SAD) experiment using the soluble domain of LjCASTOR with selenomethionine (SeMet) substitution. Incorporation of SeMet was carried out by protein expression in *E. coli* cultured in M9 medium supplemented with threonine (100 mg L$^{-1}$), lysine (100 mg L$^{-1}$), phenylalanine (100 mg L$^{-1}$), leucine (50 mg L$^{-1}$), isoleucine (50 mg L$^{-1}$), valine (50 mg L$^{-1}$), and SeMet (60 mg L$^{-1}$)[39]. The selenium-labeled protein was purified and crystallized as described above.

All diffraction data were integrated and scaled using the HKL2000 package[40]. Heavy atom positions of SeMet-substituted crystal were determined, and initial phasing and map were calculated in Phenix Autosol[41]. The structural model obtained by the SAD experiment was then employed as a search model for molecular replacement (MR), which has been performed using PHASER[42] for all other data sets including the high-resolution native data. All structural models were manually built in COOT[43] and refined in Phenix[44], and further validated using Molprobity[45]. The data collection statistics and refinement details are reported in Supplementary Table 1. All structure figures were prepared using Pymol (https://pymol.org/2/).

**Electrophysiology**. To characterize channel activity of LjCASTOR using electrophysiological studies, we generated and utilized the LjCASTOR-2TM construct where a PCR fragment spanning residues 194 and 853 was inserted into the pCGFP-Eu vector between *SalI* and *NotI* restriction sites. All single-site mutants were generated using a QuikChange Site-Directed Mutagenesis Kit (Agilent) and confirmed by DNA sequencing. About 1–2 μg of the plasmid was transfected into HEK 293 cells that were grown as a monolayer in 35-mm tissue culture dishes (to ~70% confluence) using Lipofectamine 2000 (Life Technologies). About 24–48 h after transfection, the cells were dissociated by trypsin treatment and kept in complete serum-containing medium and re-plated onto 35-mm tissue culture dishes and incubated at 37 °C until recording. Patch clamps in cell-attached, whole-cell, inside-out, and outside-out configurations were employed to measure the

LjCASTOR-2TM currents in HEK 293 cells overexpressing GFP-tagged proteins and their mutants. The standard intracellular solution contained 145 mM potassium methanesulfonate (KMS) or sodium methanesulfonate (NaMS), 5 mM KCl or NaCl, and 10 mM HEPES buffered with Tris-HCl, pH 7.4. For free $Ca^{2+}$ concentrations <100 μM, a mixture of 5 mM EGTA and a certain amount of $CaCl_2$ was prepared to achieve the target free $Ca^{2+}$ concentration according to MAXCHE-LATOR (http://maxchelator.stanford.edu). The extracellular solution contained 145 mM NaMS, 5 mM NaCl, and 10 mM HEPES buffered with Tris-HCl, pH 7.4. $Na^+$ was used as the main monovalent in most of the recordings to eliminate the effect of endogenous $K^+$ channels commonly seen in HEK cell recording. Furthermore, there is no report of endogenous channels from HEK cells that have similar gating and selectivity properties as those of LjCASTOR in this study.

The data were acquired using an AxoPatch 200B amplifier (Molecular Devices) and a low-pass analog filter set to 1 kHz. The current signal was sampled at a rate of 20 kHz using a Digidata 1322A digitizer (Molecular Devices) and further analyzed with pClamp 9 software (Molecular Devices). Patch pipettes were pulled from borosilicate glass (Harvard Apparatus) and heat polished to a resistance of 3–5 MΩ. After the patch pipette attached to the cell membrane, a gigaseal (>10 GΩ) was formed by gentle suction. The inside-out configuration was formed by pulling the pipette away from the cell, and the pipette tip was exposed into the air for a short time in some cases. The whole-cell configuration was formed by short zap or suction to rupture the patch, and pulling the pipette away from the cell after the whole-cell configuration to form the outside-out configuration. Inside-out patches were used to measure $[Ca^{2+}]$-dependent channel activation, and the single-channel open probability was measured at 100 mV of holding potential. Outside-out patches were used to measure LjCastor-2TM ion selectivity with 10 μM of free $Ca^{2+}$ in the intracellular solution for channel activation, and the single-channel currents were measured at various holding potentials from −100 mV to +100 mV to determine the reversal potential ($E_{rev}$). To measure the $Na^+$ and $K^+$ selectivity, the intracellular solution contained 145 mM NaMS, 5 mM NaCl, 10 mM HEPES buffered with Tris, pH 7.4, and 10 μM of free $Ca^{2+}$; the extracellular solution was changed to 145 mM KMS, 5 mM KCl, and 10 mM HEPES buffered with Tris-HCl, pH 7.4. To measure calcium selectivity, the extracellular solution was changed to 10 mM $Ca(MS)_2$, 5 mM $CaCl_2$, 130 mM NMDG-MS, and 10 mM HEPES buffered with Tris-HCl, pH 7.4. The ion permeability ratios were calculated with the following equations:

$$P_{Na}/P_K = [K]_o / ([Na]_i \times \exp(E_{rev}/(RT/F))),$$

$$P_{Ca}/P_{Na} = [Na]_i \times \exp(E_{rev} \times F/RT) \times (1 + \exp(E_{rev} \times F/RT)) / (4 \times [Ca]_o),$$

where $E_{rev}$ is the reverse potential measured with tail currents, $F$ is Faraday's constant, $R$ is the gas constant, and $T$ is the absolute temperature.

**$Ca^{2+}$-induced cytosolic $Ca^{2+}$ oscillation in HEK 293 cells**. General procedures for FRET-based calcium spiking assay were performed as previously described[13] with minor modifications. pIRES2-YC3.6 plasmids harboring MtDMI1 constructs were transfected using Lipofectamine 2000 (Life Technologies) into HEK 293 cells that were pre-plated on a 35-mm glass-bottomed dish (MatTek Corp.). YC3.6 is used to monitor cytosolic $Ca^{2+}$ concentration change. It consists of two fluorophores, CFP (cyan) and YFP (yellow), that are linked by calmodulin and M13 calmodulin-binding peptide, yielding a $[Ca^{2+}]$-dependent FRET between CFP and YFP[33]. About 36–48 h after transfection, the HEK 293 cells exhibiting a high level of YC3.6 expression were selected for FRET measurement. Before imaging, the complete growth medium was replaced with 2 mL of pre-warmed (37 °C) external bath solution containing 10 mM HEPES, pH 7.4, 130 mM NaCl, 3 mM KCl, 0.6 mM $MgCl_2$, and 10 mM glucose. The transfected cells were stimulated with step-wise addition of $CaCl_2$ from 0 to 2 mM, and then, 2 to 4 mM with a 5–10 min interval. Cytoplasmic calcium-dependent FRET signal was collected using a Zeiss LSM 770/880 Meta confocal microscope (Carl Zeiss Inc.). After excitation with a 405-nm laser, the CFP emission (445–493 nm) and YFP emission (517–589 nm) were recorded by scanning object freely every 7.5 s for a total of 120 cycles. The images were analyzed using Fiji image-processing package[46]. Raw data were corrected for background by performing background subtraction and then bleach correction using the exponential fit method. FRET/CFP ratio was calculated for each time point and plotted on the y-axis in a scatter diagram over time (min) on the x-axis.

**Nodulation assays with transformed legume roots**. For the wild-type and single mutants of LjCASTOR, coding sequences were synthesized by SynBioTech (Monmouth Junction, NJ). Individual constructs were transferred via GoldenGate cloning to the binary vector pAGM4673[47]. Binary vectors were designed to co-express LjCASTOR variants using the cauliflower mosaic virus 35S promoter and ER-localized red fluorescent protein (TdTomato) for screening transgenic roots. Assembled vectors were inserted into *Agrobacterium rhizogenes* MSU440 and used to generate transgenic roots. MtDMI1 mutants were created in the pDONR221 vector using the QuikChange II Site-Directed Mutagenesis Kit (Agilent). Individual MtDMI1 variants were transferred via the Gateway LR recombination reaction (Invitrogen) into the binary vector pK7FWG2-RR, which allowed co-expression of mutant constructs using the cauliflower mosaic virus 35S promoter and a red fluorescent protein (DsRed) for screening transgenic roots. Assembled vectors were

inserted into *A. rhizogenes* MSU440 and used to generate transgenic roots. *Lotus japonicus* cv. *Ljcastor-4* and *Medicago truncatula dmi1-4* (FN1) mutants were scarified by treatment with sulfuric acid for 7 min, surface-sterilized in 8% sodium hypochlorite solution for 2 min, and imbibed overnight in sterile water at 22 °C. Seeds were stratified for 4 days at 4 °C on 1.5% agar plates containing 1 μM gibberellic acid, and germinated over 5–6 days (for *L. japonicus*) or overnight (for *M. truncatula*) at 22 °C. To transform the root systems of *Ljcastor-4* or *dmi1-4* with single mutants, the roots were cut at the hypocotyls with a scalpel, and the wound was dipped into an *A. rhizogenes* MSU440 containing the binary vectors with mutant constructs. Plants were placed on 1.5% agar plates of modified Fahraeus medium supplemented with 0.5 mM $NH_4NO_3$[48]. Plants were grown under 24-h light conditions at 24 °C for 3 (for *L. japonicus*) or 2 (for *M. truncatula*) weeks until the emergence of transformed roots. For nodulation assays, the plants with their transgenic roots were transferred to 2.5″ × 2.5″ pots containing Turface® wetted with Fahraeus medium and 0.5 mM $NH_4NO_3$. The plants were watered with the same solution at first watering and with modified Fahraeus medium without $NH_4NO_3$ subsequently. Two days after potting, transgenic roots of *L. japonicus* and *M. truncatula* were inoculated with 10 mL of *M. loti* MAFF303099 diluted to an $OD_{600}$ of 0.05 in sterile water and with 10 mL of *S. meliloti* CSB357 diluted to an $OD_{600}$ of 0.01, respectively. Plants were grown under 16 h light and 8 h dark conditions at 23 °C. Transgenic roots of *L. japonicus* were harvested after 3 weeks to count nodules, and those of *M. truncatula* were collected after 2 weeks. Transgenic roots were identified using the fluorescent reporter, and the number of nodules was recorded for each transformed root.

**Reporting Summary**. Further information on research design is available in the Nature Research Reporting Summary linked to this article.

## Data availability
The atomic coordinates and structure factors have been deposited in the Protein Data Bank under accession number 6O6J for the $Ca^{2+}$-bound structure in $Na^+$, 6O7C for the $Ca^{2+}$-bound structure in $K^+$, and 6O7A for the apo structure. The source data underlying Figs. 1c, 7a–c, and 8b, c are provided as a Source Data file.

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

## Acknowledgements

We thank N. Nguyen and Z. Keyser for manuscript preparation. The experimental results reported in this article derive from the work performed at Argonne National Laboratory, Structural Biology Center (19ID) and GM/CA (23ID) at the Advanced Photon Source, and from the work performed at the Berkeley Center for Structural Biology at the Advanced Light Source (ALS). Argonne is operated by the University of Chicago Argonne, LLC, for the U.S. Department of Energy, Office of Biological and Environmental Research under contract DE-AC02-06CH11357. The Berkeley Center for Structural Biology is supported in part by the National Institutes of Health, National Institute of General Medical Sciences, and the Howard Hughes Medical Institute. The Advanced Light Source is supported by the Director, Office of Science, Office of Basic Energy Sciences, of the U.S. Department of Energy under Contract No. DE-AC02-05CH11231. This work was supported in part by the Howard Hughes Medical Institute and by grants from the National Institute of Health (GM079179 to Y.J.), the Welch Foundation (Grant I-1578 to Y.J.), and the United States Department of Agriculture AFRI (Grant #2015-67013-22899 to M.V. and J.M.A.).

## Author contributions

S.K. and J.L. performed the structure determination; W.Z. performed electrophysiology; S.K., S.B., and M.V. performed the functional assays; all authors participated in designing the research, analyzing data, and preparing the manuscript; J.A. and Y.J. supervised the project.

## Additional information

**Competing interests:** The authors declare no competing interests.

