## [Peer Review File · Nature Communications]

Reviewers' comments:

Reviewer #1 (Remarks to the Author):

The manuscript by Kim et al examines the crystal structure and function of an intracellular Ca-gated Ca channel from plants, corresponding to the DMI1 and CASTOR/POLLUX channels. Although these channels were first thought to be K channels, the authors note that the channels do not contain the canonical K channel selectivity filter sequence, and show experimentally (by single-channel recording) that the channels are selective for $\text{Ca} > \text{Na} > \text{K}$. Also, the authors find that the channels are gated by Ca in a biphasic manner, being activated with $[\text{Ca}] < 100 \mu\text{M}$ and inhibited with $[\text{Ca}] > 100 \mu\text{M}$, similar to the regulation of ER Ca channels like RYR or IP3R. Consistent with this regulation, the authors identify several binding sites for Ca and other ions within the RCK domains when they are co-crystallized with Ca. The authors then correlate these sites with function, to some extent, by several approaches, although the exact correspondence is not necessarily clear cut due to potential dual interactions of Ca-coordinating residues. The authors also present an apo structure of the RCK domains in which the four-fold symmetry is broken. This apo structure adds to the picture of conformational changes that may underlie different gating states of the channel. However, considering that the open probability is low both in high as well as low $[\text{Ca}]$, the authors suggest that both the Ca-bound and apo gating rings underlie closed conformations of the channel.

Overall the manuscript is well written, and the functional and crystallographic data are of excellent quality. This is a novel and interesting example of regulation of a Ca-selective channel by RCK domains, which were previously shown to regulate K channels and transporters, and further suggests an evolutionary relationship between RCK domain-containing K channels in prokaryotes and this potentially novel class of Ca-release channels. There are a few points in the manuscript that will need clarification.

1) The authors state that channel activation "requires cytosolic/nucleosolic Ca^{2+} ". This is a confusing; if the channels mediate release of Ca from the nucleus, where the $[\text{Ca}]$ is presumably high, then it would make sense that the RCK domains face the cytosol and sense cytosolic Ca. The authors should try to clear this up by either establishing the topology of the channel with respect to the cytosol or nucleus, or at least change the language to be more clear.

2) Based on Figure 2, the locations of the Ca binding sites in CASTOR seem similar to those in the MthK RCK domains identified in the literature. Is there any structural similarity between CASTOR and MthK in these regions? It might be of interest to include an alignment of CASTOR with the MthK RCK domain in Figure S2, and some brief discussion of this potential relationship.

Reviewer #2 (Remarks to the Author):

This is an important manuscript that starts to address the elusive role of the nuclear localized channel CASTOR in symbiotic calcium signaling. The work provides novel structural data on CASTOR and overall the structural work is of high quality. I am not an expert in electrophysiology and cannot evaluate if the claim that CASTOR functions as a calcium regulated calcium channel is conclusively supported by the functional electrophysiological data. I am also not entirely convinced that the truncated construct CASTOR-2TM missing TM1 and TM2 is a good construct reflecting the native function of the channel.

1. The crystal structures of the CASTOR soluble domain enables very detailed analyses of the structure. It would be beneficial for the generality of this study to relate the structures of CASTOR to other known channels (ligand-gated K^+ channels or others) and show their structural similarities and differences and possible implication for their ion-regulation.

2. The very high-resolution structure of CASTOR crystallized in the presence of sodium, calcium and magnesium enables a very convincing analysis of the three ion clusters, which is nicely supported by the coordination geometry and anomalous scattering data. Again it would be beneficial to compare these ion clusters to the clusters in other channels and analyze the commonalities and differences. The authors also determined the structure of CASTOR in the ion free (apo) state and show differences (mostly ridged body) between the apo and calcium bound structures. The gating ring distance does not seem to be affected, suggesting both structures are in the closed confirmation. Can this observation be rationalized also in relation to the functional data and proposed calcium regulation of CASTOR?

3. The nodulation and calcium spiking assays of different variants of CASTOR and DMI1 in Lotus and Medicago are strong confirming data that the identifies calcium sites are functional important. Additionally these experiments supports the premise that the CASTOR-2TM and DMI-2TM constructs are relevant as a proxy for the full-length versions. This point could be strengthened more, maybe with nodulation assays on the CASTOR-2TM and DMI-2TM constructs (if these are still functional). For the calcium spiking data it would be necessary to report the number of measurements (cells spiking / total cells tested) and present a good number of replicates (20 cells or so seems to be an agreed number in the literature) to convincingly show the robustness of this phenotype.

4. The authors argues against the potassium channel model (line 90-95). It would be useful with a clear reference to the original study(s) the authors point to here. The authors claim that DMI1 and CASTOR was denatured and not correctly folded when reconstituted into proteoliposomes (line 90-95) in these study(s). The authors should provide data supporting this conclusion.

6. Figure S2 has some formatting errors and are missing protein IDs. It would also be useful to indicate on this figure the constructs used for the functional assays (also indicate mutations) and structural studies. If possible also indicate the series of N-terminal CASTOR constructs generated that failed to localize at the surface of HEK293 cells as these negative result are often left out and could help future designs of these type of experiments.

7. Generally the experiments are well describe and with sufficient details to allow reproduction. One missing detail is the gel filtration buffer used for purifying both the apo and ligand bound state of CASTOR and clearly stating the final concentration of ions in the crystallization conditions and during cryo-protection.

Reviewer #3 (Remarks to the Author):

In this work, Kim et al use *L. japonicus* CASTOR as a model system to demonstrate that the plant DMI1, POLLUX and CASTOR nuclear ion channels that are responsible for root symbiosis with rhizobia bacteria, operate as tetrameric Ca²⁺-regulated Ca channels whose ligand binding domains form an RCK gating ring. The CASTOR exhibits high Ca²⁺ selectivity over K⁺ or Na⁺. This is in a contrast with the previous suggestion that these channels function as K⁺ channels. The paper is a mix of electrophysiological studies conducted using heterologous HEK293 system and structural studies involving crystallization of CASTOR soluble domain.

While my overall impression of this work is generally positive, there are a few issues that make me uncomfortable. My major concerns are as follows:

1. The major bulk of measurements was conducted in artificial heterologous systems; this is true for all electrophysiological studies and for FRET-based Ca oscillations assays. I would treat this data as preliminary but would trust the conclusions only seeing this channel operating in planta. Given that the authors have already generated transgenic *M. truncatula* and *L. japonicus* plants, as

a very least they should demonstrate that CASTOR-mediated Ca oscillations are present in some but absent in other point mutants.

2. The functional characterization of LjCASTOR was conducted in HEC system using truncated 2TM sequence as, according to the authors, patch-clamp experiments on the PM of HEK cells expressing full-length CASTOR was not successful due to the lack of surface expression of these channels (In 111-114). I have a very major reservation about this approach. As commented earlier (In 68-72), a single amino acid substitution Ser-to-Ala is responsible for the functional difference between DMI1 and CASTOR channels. Here, the authors omit a substantial chunk of the channel (not just one AA); how can they be certain that the functional properties will be unchanged and comparable with those in planta? I accept the authors point that such measurements are not possible in HEK system but via can't they use another heterologous expression system (e.g. *Xenopus oocytes*?). Without such measurements, the entire story is rather shaky.

3. The authors implicate CASTOR channel as a component of CACR mechanism responsible for generating Ca spikes required for nodulation. The data shown in Fig 7A support this conclusion, but only partially. Given that in HEK cells CASTOR channels are expressed at the plasma membrane, how can the authors be sure that in plant this channel is located exclusively at the nuclear envelope? What if a small portion of CASTOR channels is present at the PM, and they modulate Ca uptake from the apoplastic space? Evidence for channel's localization must be provided.

4. I am also slightly lost with the physiological relevance of the reported bell-shaped Ca-activation curve. The range of concentration used (30 μ M to 1 mM) is non-physiological. Smaller (including sub-micromolar) concentrations should be used.

Reviewer #4 (Remarks to the Author):

The discoveries in this manuscript by Kim et al. have been long awaited in the field of plant ion channels, which are involved in calcium signaling mechanisms responsible for abiotic and biotic stress perception and responses. It is well known that homologous DMI1, CASTOR and POLLUX ion channels play a key role in oscillations of the nuclear and perinuclear Ca²⁺ concentration, which are essential for symbiotic associations between legume plants and rhizobial bacteria or mycorrhizal fungi. Excitingly, in this study the breakthrough of expression a truncated LjCASTOR (LjCASTOR-2TM) in the plasma membrane in HEK cells made it possible to analyze the biophysical properties of LjCASTOR in an exogenous expression system. Several of their findings are fundamentally important and concept setting, and will have broader impacts on calcium signaling in the symbiotic associations in small and in abiotic and biotic stress perception in large. LjCASTOR-2TM forms calcium permeable channels and is up- and down-regulated by calcium, meaning that DMI1, CASTOR and POLLUX ion channels themselves are possibly sufficient from the symbiotic calcium oscillations via feedforward and feedback mechanisms. In contrast, previous studies show that DMI1, CASTOR and POLLUX are K⁺ channels, and CNGCs are the Ca²⁺ channels mediating the symbiotic calcium oscillations. Even more exciting, they have solved the 3D structures of the soluble portion with RCK domains, and identified and tested the binding sites for Ca²⁺ among other ions with impressive amount of site specific mutations. These binding sites for Ca²⁺ are crucial for the channel structure as well as gating. Finally, they extended their findings from LjCASTOR to MtDMI1 and tested these Ca²⁺ binding sites in nodule formation in *M. truncatula* and calcium oscillations in HEK cells. The data are enormous, the findings are novel, and the conclusions are important and physiological relevant. Certainly, it would be interesting to know the pore structure with an ion selective filter of the channel in the future. The manuscript is very suitable for publication in Nature Communications after addressing a few minor concerns listed below.

1. The LjCASTOR-2TM currents were analyzed at the single channel level not at the whole-cell level. Considering there are a few endogenous cation conducting channels in HEK cells, and given that the LjCASTOR-2TM currents were small, these important channel properties drawn from these single channel analyses, such as calcium activation and ion selectivity of Ca²⁺ over K⁺, might not be very accurate. In addition, key controls (such as empty vector-transfected cells) were missing, which makes the referee hard to evaluate these single channel currents.

--- Fig. 1. "...of HEK 293 cells expressing full-length *L. japonicus* CASTOR (LjCASTOR) was not successful due to the lack of surface expression of the channel and the majority of the expressed proteins being localized in the ER and nuclear membranes". Data should be shown.

--- "Fig. 1A shows the whole cell current in the presence of symmetrical 150 mM Na⁺, 120 illustrating the cationic (Na⁺) currents conducted by LjCASTOR-2TM". The control whole-cell currents in HEK cells transfected with the empty vector should be shown.

--- HEK cell endogenous cation channels should be discussed.

--- For the I-V curves in Fig. 1B, D, E, it seems that only one recording was plotted. Considering the importance of the selectivity, adequate amount of recordings should be performed and analyzed.

2. The ion selectivity of LjCASTOR-2TM currents. From single channel analyses of the wild type and mutants of LjCASTOR-2TM, these authors conclude that LjCASTOR-2TM currents are highly selective for Ca²⁺ over K⁺, which differs from previous studies. LjCASTOR-2TM currents in the whole-cell mode or cell-attached macroscopic currents should be analyzed with substitution of cations, and calcium blockers such as La³⁺ should be applied.

3. The role of DMI1, POLLUX and CASTOR vs CNGCs in conducting calcium ions in the symbiotic associations should be discussed. CNGC channels have Ca²⁺/CAM binding domains, and could be potentially regulated by Ca²⁺ to generate oscillations. Previous studies propose that DMI1, POLLUX and CASTOR channels are K⁺ channels, which facilitate or regulate indirectly Ca²⁺ channels for symbiotic calcium oscillations, while CNGCs are these Ca²⁺ channels. These studies should be cited and discussed, especially the Ca²⁺ binding sites for channel gating.

4. Progress in structural studies on plant Ca²⁺ channels should be discussed to promote the novelty and importance of the current LjCASTOR study. In contrasted to numerous animal Ca²⁺ channels, such as TRP and Piezo, the genes for calcium machinery, especially for Ca²⁺ channels responsible for biotic and abiotic stress perception, remain largely to be identified in plants. The structure information for plant Ca²⁺ channels is essentially lacking. Thus, this study on LjCASTOR will be highly appreciated in the field. Recently, the whole structures of osmosensor OSCA channels have been solved, which are published in Nature Structural & Molecular Biology, Nature Communication, and eLife. These studies should be cited. OSCAs could sense osmotic/mechanical stresses and mediate Ca²⁺ influx, and whether they play a role in the initial Ca²⁺ influx in symbiotic associations may be discussed.

To referee #1:

We want to thank the reviewer for the positive critique of our manuscript. The reviewer has provided some constructive suggestions, and the points raised by the reviewer are addressed as follow.

1) The authors state that channel activation “requires cytosolic/nucleosolic Ca²⁺.” This is a confusing; if the channels mediate release of Ca from the nucleus, where the [Ca] is presumably high, then it would make sense that the RCK domains face the cytosol and sense cytosolic Ca. The authors should try to clear this up by either establishing the topology of the channel with respect to the cytosol or nucleus, or at least change the language to be more clear.

Studies on DMI1 indicate that CASTOR/POLLUX/DMI1 channels are localized in both the outer and the inner membranes of the nuclear envelope. Therefore, their Ca²⁺-binding RCK domains would face the cytosol when the channels are in the outer membrane and nucleosol when the channels are in the inner membrane. These channels likely mediate the Ca²⁺ release from the nuclear envelope (which is continuous with the endoplasmic reticulum) to nucleosol as well as cytosol. We have included the reference about our previous DMI1 localization study (Capoen *et al.*, 2011) in the revision and have also revised the text to clarify.

2) Based on Figure 2, the locations of the Ca binding sites in CASTOR seem similar to those in the MthK RCK domains identified in the literature. Is there any structural similarity between CASTOR and MthK in these regions? It might be of interest to include an alignment of CASTOR with the MthK RCK domain in Figure S2, and some brief discussion of this potential relationship.

The Ca²⁺ binding sites in CASTOR are quite different from those in MthK and BK channels. Although overall the structures are conserved among RCK domains, there is no sequence conservation between the CASTOR family of channels and other RCK containing K⁺ channels. For this reason, we did not include an alignment of CASTOR with the MthK or BK RCK domains.

To referee #2:

We want to thank the reviewer for the positive critique of our manuscript. The reviewer has provided some constructive suggestions, and the points raised by the reviewer are addressed as follow.

This is an important manuscript that starts to address the elusive role of the nuclear localized channel CASTOR in symbiotic calcium signaling. The work provides novel structural data on CASTOR and overall the structural work is of high quality. I am not an expert in electrophysiology and cannot evaluate if the calm that CASTOR functions as a calcium regulated

calcium channel is conclusively supported by the functional electrophysiological data. I am also not entirely convinced that the truncated construct CASTOR-2TM missing TM1 and TM2 is a good constructs reflecting the native function of the channel.

Like other tetrameric cation channels, the last two transmembrane domains (TM) of CASTOR form the ion conduction pore that defines the selectivity and gating of the channel. Although we cannot rule out the possibility that the first two TMs may play some regulatory role such as protein localization, we are confident that the 2-TM construct in our study provides a sound model system for characterizing the biophysical properties (selectivity and gating) of the CASTOR channel. In the Ca^{2+} spiking assay, we have tested both the full-length channel as well as the 2-TM construct and observed similar Ca^{2+} spiking patterns.

1. The crystal structures of the CASTOR soluble domain enables very detailed analyses of the structure. It would be beneficial for the generality of this study to relate the structures of CASTOR to other known channels (ligand-gated K^+ channels or others) and show their structural similarities and differences and possible implication for their ion-regulation.

While the overall structures are conserved among RCK domains, the Ca^{2+} binding sites in CASTOR are quite different from those in ligand-gated K^+ channels such as MthK and BK channels. In the revision, we have included a supplementary figure comparing the overall structure of the RCK gating rings from CASTOR, MthK and BK channels and their respective Ca^{2+} binding sites.

2. The very high-resolution structure of CASTOR crystallized in the presence of sodium, calcium, and magnesium enables a very convincing analysis of the three ion clusters, which is nicely supported by the coordination geometry and anomalous scattering data. Again it would be beneficial to compare these ion clusters to the clusters in other channels and analyze the commonalities and differences. The authors also determined the structure of CASTOR in the ion free (apo) state and show differences (mostly ridged body) between the apo and calcium bound structures. The gating ring distance does not seem to be affected, suggesting both structures are in the closed confirmation. Can this observation be rationalized also in relation to the functional data and proposed calcium regulation of CASTOR?

Although the overall structures are conserved among RCK domains, there is no sequence conservation between CASTOR/POLLUX/DMI1 channels and other RCK-containing K^+ channels, and the Ca^{2+} binding sites in CASTOR are also different from those in MthK and BK channels. This divergence in Ca^{2+} binding among these RCK-regulated channels makes it difficult to compare and generalize their ion-binding mechanism.

Our electrophysiology data demonstrated that the CASTOR channel has a low single channel open probability even in the presence of Ca^{2+} , suggesting that the channel is stable in the closed state for most of the time even with ligand bound.

3. The nodulation and calcium spiking assays of different variants of CASTOR and DMI1 in Lotus and Medicago are strong confirming data that the identifies calcium sites are functional

important. Additionally these experiments supports the premise that the CASTOR-2TM and DMI-2TM constructs are relevant as a proxy for the full-length versions. This point could be strengthened more, maybe with nodulation assays on the CASTOR-2TM and DMI-2TM constructs (if these are still functional). For the calcium spiking data it would be necessary to report the number of measurements (cells spiking / total cells tested) and present a good number of replicates (20 cells or so seems to be an agreed number in the literature) to convincingly show the robustness of this phenotype.

In a previous study (Riely *et al.*, 2006), we showed that the expression of DMI1 mutant with its N-terminal soluble region removed failed to form root nodules in the nodulation assays, as the soluble N-terminus is required for channel localization (<https://onlinelibrary.wiley.com/doi/full/10.1111/j.1365-313X.2006.02957.x>). With the N-terminus and first two TMs missing, we do not expect the 2-TM constructs of DMI1 and CASTOR will work in the nodulation assays.

As suggested, we have added statistical analyses to the Ca²⁺ spiking assay in the revision.

4. The authors argues against the potassium channel model (line 90-95). It would be useful with a clear reference to the original study(s) the authors point to here. The authors claim that DMI1 and CASTOR was denatured and not correctly folded when reconstituted into proteoliposomes (line 90-95) in these study(s). The authors should provide data supporting this conclusion.

The references related to the K⁺ channel model have been added in the revision. As described in the methods of these studies, the CASTOR protein was solubilized in SDS and denatured in 8 M urea before being used in the reconstitution of the proteoliposomes.

5. Figure S2 has some formatting errors and are missing protein IDs. It would also be useful to indicate on this figure the constructs used for the functional assays (also indicate mutations) and structural studies. If possible also indicate the series of N-terminal CASTOR constructs generated that failed to localize at the surface of HEK293 cells as these negative result are often left out and could help future designs of these type of experiments.

Thanks for the suggestions. Corrections have been made to Figure S2 as suggested in the revision.

6. Generally the experiments are well describe and with sufficient details to allow reproduction. One missing detail is the gel filtration buffer used for purifying both the apo and ligand bound state of CASTOR and clearly stating the final concentration of ions in the crystallization conditions and during cryo-protection.

Thanks for the suggestions. We have included the requested information in the Methods section.

To referee #3:

We thank the reviewer for the positive comments and constructive suggestions. The major points raised by the reviewer are addressed as follow.

1. The major bulk of measurements was conducted in artificial heterologous systems; this is true for all electrophysiological studies and for FRET-based Ca oscillations assays. I would treat this data as preliminary but would trust the conclusions only seeing this channel operating in planta. Given that the authors have already generated transgenic *M. truncatula* and *L. japonicus* plants, as a very least they should demonstrate that CASTOR-mediated Ca oscillations are present in some but absent in other point mutants.

Ca²⁺ measurements in plants are not a reliable assay to evaluate DMI1, CASTOR and POLLUX activity. Ca²⁺ spiking frequency and amplitude are, unfortunately, too variable from cell to cell to correlate them to channel activity. Several publications used the number of nodules as a reliable assay to evaluate the activity of DMI1, CASTOR, POLLUX wild-type and mutants in planta:

- in *Medicago truncatula*, Venkateshwaran *et al.* 2012:

<http://www.plantcell.org/content/24/6/2528.long>

- in *Lotus japonicus*, Charpentier *et al.* 2008:

<http://www.plantcell.org/content/20/12/3467>

2. The functional characterization of LjCASTOR was conducted in HEC system using truncated 2TM sequence as, according to the authors, patch-clamp experiments on the PM of HEK cells expressing full-length CASTOR was not successful due to the lack of surface expression of these channels (ln 111-114). I have a very major reservation about this approach. As commented earlier (ln 68-72), a single amino acid substitution Ser-to-Ala is responsible for the functional difference between DMI1 and CASTOR channels. Here, the authors omit a substantial chunk of the channel (not just one AA); how can they be certain that the functional properties will be unchanged and comparable with those in planta? I accept the authors point that such measurements are not possible in HEK system but via can't they use another heterologous expression system (e.g. *Xenopus* oocytes?). Without such measurements, the entire story is rather shaky.

We have spent over three years trying to establish the electrophysiology recordings of CASTOR and DMI1. We have tested various protein constructs in different heterologous expression systems including HEK293 and *Xenopus* oocytes. We have also tried to reconstitute the channel proteins into proteoliposomes for channel electrophysiology using liposome patch and lipid bilayer. So far, we can only record channel activity using the 2-TM constructs of CASTOR and DMI1 expressed in HEK cells.

Like the other tetrameric cation channels, the last two TMs (TM3 and 4) of CASTOR form the ion conduction pore that defines the selectivity and gating of the channel. Although we cannot rule out the possibility that the first two TMs may play some regulatory role such as protein localization, we are confident that the 2TM construct provides a sound model system to characterize the biophysical properties (selectivity and gating) of the CASTOR channel. In the Ca²⁺ spiking assay, we tested both the full-length channel and the 2-TM

construct and observed similar Ca^{2+} spiking patterns (Figure 7c). The Ser-to-Ala substitution between CASTOR and DMI1 occurs at the selectivity filter region of the ion conduction pore that is formed by the last two TMs. This residue difference between the two channels likely contributes to the difference in channel conductance.

3. The authors implicate CASTOR channel as a component of CACR mechanism responsible for generating Ca spikes required for nodulation. The data shown in Fig 7A support this conclusion, but only partially. Given that in HEK cells CASTOR channels are expressed at the plasma membrane, how can the authors be sure that in plant this channel is located exclusively at the nuclear envelope? What if a small portion of CASTOR channels is present at the PM, and they modulate Ca uptake from the apoplastic space? Evidence for channel's localization must be provided.

The localization of DMI1/CASTOR to the nuclear membrane has been demonstrated in several previous studies, which are listed below and referenced in our manuscript. When overexpressed in HEK cells, DMI1 and CASTOR are predominantly localized to the ER and nuclear membranes.

<https://www.pnas.org/content/108/34/14348>

<https://onlinelibrary.wiley.com/doi/full/10.1111/j.1365-313X.2006.02957.x>

<http://www.plantcell.org/content/20/12/3467>

4. I am also slightly lost with the physiological relevance of the reported bell-shaped Ca-activation curve. The range of concentration used (30 μM to 1 mM) is non-physiological. Smaller (including sub-micromolar) concentrations should be used.

CASTOR has very low channel activity at sub-micromolar Ca^{2+} concentrations. The concentration range of Ca^{2+} that yields bell-shaped Ca^{2+} activation curve is comparable to that observed in RyR channels. When Ca^{2+} conducting channels release Ca^{2+} into the cytosol (i.e., from ER/SR), the global cytosolic Ca^{2+} peaks at around 1 μM ; however, the regions close to the open Ca^{2+} channels can achieve much higher Ca^{2+} concentrations. These local high Ca^{2+} concentration “microdomains” can provide a meaningful signal for low-affinity Ca^{2+} -sensing motifs that are unresponsive to fluctuations in the global cytosolic Ca^{2+} concentration.

To referee #4:

We want to thank the reviewer for the positive critique of our manuscript. The reviewer has provided some constructive suggestions, and the few minor concerns raised by the reviewer are addressed as follow.

1. The LjCASTOR-2TM currents were analyzed at the single channel level not at the whole-cell level. Considering there are a few endogenous cation conducting channels in HEK cells, and given that the LjCASTOR-2TM currents were small, these important channel properties drawn from these single channel analyses, such as calcium activation

and ion selectivity of Ca²⁺ over K⁺, might not be very accurate. In addition, key controls (such as empty vector-transfected cells) were missing, which makes the referee hard to evaluate these single channel currents.

Control recording of cells transfected with empty vector has been included in the revision as suggested. We used Na⁺ in most of the recordings to eliminate the effect of endogenous K⁺ channels commonly seen in HEK cell recording. We have been using HEK cell recording for many channels and have never observed any endogenous channel with activity similar to that of CASTOR. With a high single-channel conductance, the single channel recordings of CASTOR actually provide a better measurement of the selectivity and gating properties of the channel.

--- Fig. 1. “..of HEK 293 cells expressing full-length L. japonicus CASTOR (LjCASTOR) was not successful due to the lack of surface expression of the channel and the majority of the expressed proteins being localized in the ER and nuclear membranes”. Data should be shown.

The localization of CASTOR/DMI1 expressed in HEK cells is shown in Supplementary Figure 10.

--- “Fig. 1A shows the whole cell current in the presence of symmetrical 150 mM Na⁺, 120 illustrating the cationic (Na⁺) currents conducted by LjCASTOR-2TM”. The control whole-cell currents in HEK cells transfected with the empty vector should be shown.

The control whole-cell currents in HEK cells transfected with the empty vector have been added in the revision as suggested.

--- HEK cell endogenous cation channels should be discussed.

As we discussed above, we used Na⁺ in most of recordings to eliminate the effect of endogenous K⁺ channels commonly seen in HEK cell recording. We have never observed any endogenous channel with activity similar to that of CASTOR.

--- For the I-V curves in Fig. 1B, D, E, it seems that only one recording was plotted. Considering the importance of the selectivity, adequate amount of recordings should be performed and analyzed.

Fig. 1B, D, E are the sample I-V curves from one experiment. All measurements have been performed at least five times with consistent results. We have included statistical analyses for the reversal potentials from multiple measurements in the revised manuscript.

2. The ion selectivity of LjCASTOR-2TM currents. From single channel analyses of the wild

type and mutants of LjCASTOR-2TM, these authors conclude that LjCASTOR-2TM currents are highly selective for Ca²⁺ over K⁺, which differs from previous studies. LjCASTOR-2TM currents in the whole-cell mode or cell-attached macroscopic currents should be analyzed with substitution of cations, and calcium blockers such as La³⁺ should be applied.

The selectivity of CASTOR was measured using outside-out patches in bi-ionic conditions with Na⁺ in the pipette and various cations (Na⁺, K⁺ or Ca²⁺) in the bath solution. This single channel recording allows for better control of the ionic components on both sides of the channel and provides a more accurate measurement of the reversal potentials and ion selectivity. The cell attached measurement, on the other hand, will not allow us to change the cytosolic solutions. La³⁺ also can block LjCASTOR-2TM from the extracellular side and we have included the data in the revision.

3. The role of DMI1, POLLUX and CASTOR vs CNGCs in conducting calcium ions in the symbiotic associations should be discussed. CNGC channels have Ca²⁺/CAM binding domains, and could be potentially regulated by Ca²⁺ to generate oscillations. Previous studies propose that DMI1, POLLUX and CASTOR channels are K⁺ channels, which facilitate or regulate indirectly Ca²⁺ channels for symbiotic calcium oscillations, while CNGCs are these Ca²⁺ channels. These studies should be cited and discussed, especially the Ca²⁺ binding sites for channel gating.

The recent study on CNGC15s and its role in the symbiotic associations was briefly discussed in the Discussion section. However, the interplay between CNGC15s and DMI1/CASTOR/POLLUX for symbiotic calcium oscillations is unclear and requires further study. Another recent study demonstrated that some CNGC channels (CNGC18, CNGC8, and CNGC7) can be regulated by Ca²⁺/CaM binding and control cytosolic calcium oscillations :

<https://www.sciencedirect.com/science/article/pii/S1534580718311250?via%3Dihub>).

However, these CNGC channels appear to be different from CNGC15s. As there is no evidence to suggest that Ca²⁺/CaM regulates CNGC15s, we are not in the position to speculate about the potential role of Ca²⁺/CaM in symbiotic calcium oscillations.

4. Progress in structural studies on plant Ca²⁺ channels should be discussed to promote the novelty and importance of the current LjCASTOR study. In contrasted to numerous animal Ca²⁺ channels, such as TRP and Piezo, the genes for calcium machinery, especially for Ca²⁺ channels responsible for biotic and abiotic stress perception, remain largely to be identified in plants. The structure information for plant Ca²⁺ channels is essentially lacking. Thus, this study on LjCASTOR will be highly appreciated in the field. Recently, the whole structures of osmosensor OSCA channels have been solved, which are published in Nature Structural & Molecular Biology, Nature Communication, and eLife. These studies should be cited. OSCAs could sense osmotic/mechanical stresses and mediate Ca²⁺ influx, and whether they play a role in the initial Ca²⁺ influx in symbiotic associations may be discussed.

Our current structure represents only the soluble, ligand-binding region of CASTOR and does not include the transmembrane pore. We are hesitant to claim that we have a plant Ca²⁺

channel structure. Connections between the mechanosensitive OSCA channels and our current study are limited. We feel that a general discussion about the structural studies of plant Ca^{2+} channels as well as the possible role of OSCA channels in the initial Ca^{2+} influx in symbiotic associations would be a bit farfetched.

REVIEWERS' COMMENTS:

Reviewer #1 (Remarks to the Author):

The revised manuscript by Kim et al examines the crystal structure and function of an intracellular Ca-gated Ca channel from plants, corresponding to the DMI1 and CASTOR/POLLUX channels. My concerns raised on the previous version of this manuscript were addressed adequately, and I have no further comments.

Reviewer #3 (Remarks to the Author):

The authors did a good job addressing concerns raised by all reviewers. My only serious issue was the question of whether truncated construct CASTOR-2TM missing TM1 and TM2 is a good construct reflecting the native function of the channel. This concern was also echoed by Reviewer#2. However, I am happy to accept the authors' explanations that in-planta Ca²⁺ spiking patterns are similar between plants harboring truncated and full-length copies; also convincing are the arguments about the location of the selectivity filter. Thus, I am satisfied with the extent of revision and have no more critical comments about this work.